# Inbreeding in a dioecious plant has sex- and population origin-specific effects on its interactions with pollinators

**Karin Schrieber**[1]*, **Sarah Catherine Paul**[2], **Levke Valena Höche**[1], **Andrea Cecilia Salas**[1], **Rabi Didszun**[1], **Jakob Mößnang**[1], **Caroline Müller**[2], **Alexandra Erfmeier**[1,3], **Elisabeth Johanna Eilers**[2]

[1]Kiel University, Institute for Ecosystem Research, Geobotany, Kiel, Germany; [2]Bielefeld University, Faculty of Biology, Department of Chemical Ecology, Bielefeld, Germany; [3]German Centre for Integrative Biodiversity Research (iDiv) Halle–Jena–Leipzig, Leipzig, Germany

**Abstract** We study the effects of inbreeding in a dioecious plant on its interaction with pollinating insects and test whether the magnitude of such effects is shaped by plant individual sex and the evolutionary histories of plant populations. We recorded spatial, scent, colour, and rewarding flower traits as well as pollinator visitation rates in experimentally inbred and outbred, male and female *Silene latifolia* plants from European and North American populations differing in their evolutionary histories. We found that inbreeding specifically impairs spatial flower traits and floral scent. Our results support that sex-specific selection and gene expression may have partially magnified these inbreeding costs for females, and that divergent evolutionary histories altered the genetic architecture underlying inbreeding effects across population origins. Moreover, the results indicate that inbreeding effects on floral scent may have a huge potential to disrupt interactions among plants and nocturnal moth pollinators, which are mediated by elaborate chemical communication.

*For correspondence:
kschrieber@ecology.uni-kiel.de

## Introduction

Plant-pollinator interactions are of central importance for the emergence as well as the maintenance of global biodiversity (*Crepet and Niklas, 2009*; *Ollerton, 2017*) and provide ecosystem services with tangible sociocultural and economic value (*Gill et al., 2016*; *Porto et al., 2020*). Global change continues to disrupt these interactions by altering the physiology, phenology, and particularly the spatial distribution of component species (*Burkle et al., 2013*; *Vanbergen, 2013*; *Glenny et al., 2018*). Habitat degradation and fragmentation reduce the size and connectivity of plant populations, which results in lowered pollinator visitation rates (*Aguilar et al., 2006*; *Dauber et al., 2010*). Plant population retraction and isolation may also affect interactions with pollinators at the plant individual level by increasing inbreeding rates (*Carr et al., 2014*). The mating among closely related plant individuals may compromise floral traits attracting pollinators and hence cause negative feedback on pollinator visitation. Mechanistic insight into the effects of inbreeding on plant-pollinator interactions and intrinsic factors shaping the magnitude of such effects is limited but urgently required for the conservation of component species.

Inbreeding increases homozygosity in the offspring generation. This may enhance the phenotypic expression of deleterious recessive mutations (i.e., dominance) and reduce heterozygote advantage (i.e., over-dominance), which can result in severe declines of Darwinian fitness in inbred relative to outcrossed offspring (i.e., inbreeding depression) (*Charlesworth and Willis, 2009*). Inbreeding may in addition disrupt plant-insect interactions. While it is well established that inbreeding can increase

**eLife digest** Destroying habitats can reduce the size of local populations of many plants and animals. For plants, a smaller population means a greater chance of inbreeding, where individual plants that are closely related to each other mate and produce offspring. Inbreeding often results in offspring that are weaker than their parents which can reduce the plant's chance of survival.

Many plants rely on animals to help them to breed. For example, bees carry pollen – containing the male sex cell – to other flowers which then fertilize the plant to produce seeds. Flowers use a wide range of attributes to attract animals such as their colour, scent and providing them with food. However, inbreeding may alter these characteristics which could make it harder for inbred plants to reproduce, meaning that populations would end up shrinking even faster.

To test this theory, Schrieber et al. studied flowers from white campions which use moths to breed. Inbred plants had smaller and fewer flowers, and had a different smell. In particular, they produced less of a chemical scent that is known to attract moths at night. Schrieber et al. then tracked moths visiting a mixed population of inbred and control plants. Fewer moths visited the inbred flowers, particularly the ones that were female. This shows that inbreeding may accelerate population loss and extinction by making flowers less attractive to animals.

This work highlights the impact habitat destruction has on plants and shows how species can decline rapidly as populations shrink. This could help to support conservation efforts and inform ecology models to better understand our effect on the environment.

a plant's susceptibility to herbivores by diminishing morphological and chemical defences (*Campbell et al., 2013*; *Kariyat et al., 2012*; *Kalske et al., 2014*), its effects on plant-pollinator interactions are less well understood. Inbreeding may reduce a plant's attractiveness to pollinating insects by compromising the complex set of floral traits involved in interspecific communication. These traits comprise (i) the spatial arrangement of individual flowers (e.g., size, shape) and multiple flowers within an inflorescence (e.g., number, height above ground, degree of aggregation), herein-after referred to as spatial flower traits (*Dafni et al., 1997*); (ii) the scent bouquet as determined by the composition of floral volatile organic compounds (VOC) such as terpenoids, benzenoids, and phenylpropanoids (*Muhlemann et al., 2014*; *Borghi et al., 2017*); (iii) flower colour as defined by the composition of pigments with wavelength-selective light absorption and the backscattering of light by petal surface structures (*van der Kooi et al., 2016*; *Borghi et al., 2017*); and (iv) the quality and quantity of rewards such as nectar, pollen, oviposition sites, or shelter (*Simpson and Neff, 1981*). These cues are particularly efficient in attracting pollinators across either long, medium, or short distances and act synergistically in determining visitation rates (*Dafni et al., 1997*; *Muhlemann et al., 2014*) . Although in a few cases inbreeding has been shown to alter single floral traits (*Ivey and Carr, 2005*; *Ferrari et al., 2006*; *Haber et al., 2019*), insight into more syndrome-wide effects is restricted to a single study. *Kariyat et al., 2021* demonstrated that inbred *Solanum carolinense* L. display reduced flower size, pollen and scent production, and receive fewer visits from diurnal generalists. It is necessary to broaden such integrated methodological approaches to other plant-pollinator systems (e.g., nocturnal specialist pollinators) and further floral traits (e.g., flower colour).

The magnitude and slope of inbreeding effects in plants can vary across environments, since local conditions partly determine the selective value of recessive alleles unmasked by inbreeding (*Fox and Reed, 2011*). While the influence of environmental stress on the expression of inbreeding depression is well studied, the effects of plant sex, which considerably shapes an individual's interaction with its environment, remain largely unexplored. Individuals of dioecious plant species invest into either male or female reproductive function. This partitioning goes along with different life histories, resource demands, stress susceptibilities, and consequently sex-specific selection regimes in identical habitats (*Moore and Pannell, 2011*; *Barrett and Hough, 2013*). Sex-specific selection may modify the magnitude of inbreeding depression in dioecious plants. Studies on animals reported higher inbreeding depression in females than males resulting from higher reproductive investment and prolonged life cycles in the former (*Ebel and Phillips, 2016*). In plants, such relations have rarely been investigated (*Teixeira et al., 2009*), and if so, not with a focus on floral traits. If inbreeding

effects on floral traits are more pronounced in female than male plants, the relative frequency of pollinator visits may be biased towards the latter sex, with devastating consequences for the effective size and persistence of populations. Studies on sex-specific inbreeding effects on floral traits are thus needed to improve the risk assessment for the conservation of dioecious plant species.

Plant populations may escape progressive retractions under increased inbreeding rates by purging. Inbreeding unmasks deleterious recessive mutations, which facilitates their selective removal from the population gene pool and may result in a rebound of fitness when the demographic bottleneck is intermediate (*Crnokrak and Barrett, 2002*). Plant species that have successfully colonised distant geographic regions provide perfect models for studying the relevance of purging in natural plant populations. As colonisation events are associated with successive demographic bottlenecks, purging is expected to be one determinant for the successful establishment and proliferation of plant populations in novel habitats (*Facon et al., 2011*; *Schrieber and Lachmuth, 2017*). However, only few empirical studies verified the role of purging in plant colonisation success by revealing significantly lower inbreeding depression following experimental crossings in invasive than native plant populations (*Rosche et al., 2016*; *Schrieber et al., 2019b*). Again, the focus of these studies was on fitness components rather than floral traits. Yet, attractiveness to pollinators is a key for successful colonisation in species introduced into novel communities (*Morales and Traveset, 2009*), especially if plants are not capable of selfing (e.g., dioecious). Floral syndromes are likely under strong selection in such species, which should rapidly purge deleterious recessive mutations affecting spatial, scent, colour, and rewarding flower traits.

In the present study, we investigated the effects of inbreeding on plant-pollinator interactions and tested whether the magnitude of such effects depends on plant sex and population origin using the *Silene latifolia* PORR. (Caryophyllaceae) and its crepuscular moth pollinators. Natural *S. latifolia* populations partly suffer from biparental inbreeding due to limited seed and pollen dispersal (*McCauley, 1997*). As inbreeding reduces not only fitness (*Teixeira et al., 2009*) but also impairs interactions with herbivorous insects in *S. latifolia* (*Schrieber et al., 2019a*; *Schrieber et al., 2019b*), a disruption of plant-pollinator interactions can be expected for this species. Moreover, the dioecious reproductive system of *S. latifolia* provides the opportunity to quantify variation in the magnitude of inbreeding effects in these traits among females and males. Finally, the species expanded successfully from parts of its native distribution range in Europe to North America in the early 19th century (*Keller et al., 2009*; *Keller et al., 2012*), which may have given rise to purging events. We assessed spatial flower traits, headspace floral scent composition, flower colour and rewards, and quantified pollinator visitation in experimentally inbred and outbred male and female *S. latifolia* individuals from European and North American populations. We hypothesised that (i) inbreeding compromises floral traits, (ii) these inbreeding effects are more pronounced in female than male plants, (iii) inbreeding effects are more pronounced in European than North American populations, and (iv) the combined effects of inbreeding, sex, and population origin cause feedback on pollinator visitation rates.

## Materials and methods

### Study species

*S. latifolia* shows a distinct moth pollination syndrome with large, white, and funnel-shaped flowers (*Dafni et al., 1997*). The flowers open from dusk till mid-morning to release a scent bouquet composed of more than 60 VOC, whereby emission peaks around dusk (*Dötterl and Jürgens, 2005*; *Dötterl et al., 2009*; *Mamadalieva et al., 2014*). During the daytime, no measurable floral scent is emitted (*Dötterl et al., 2005*). Nectar production peaks 3–4 days after flower opening and is just as floral scent emission reduced after pollination (*Gehring et al., 2004*; *Dötterl and Jürgens, 2005*; *Muhlemann et al., 2006*). *S. latifolia* exhibits various sexual dimorphisms with male plants producing more and smaller flowers that excrete lower volumes of nectar with higher sugar concentrations as compared to females (*Gehring et al., 2004*; *Delph et al., 2010*). The quality of floral scent exhibits no clear sex-specific patterns, while male plants have been shown to emit higher or equal total amounts of VOC as compared to females in different studies (*Dötterl and Jürgens, 2005*; *Waelti et al., 2009*).

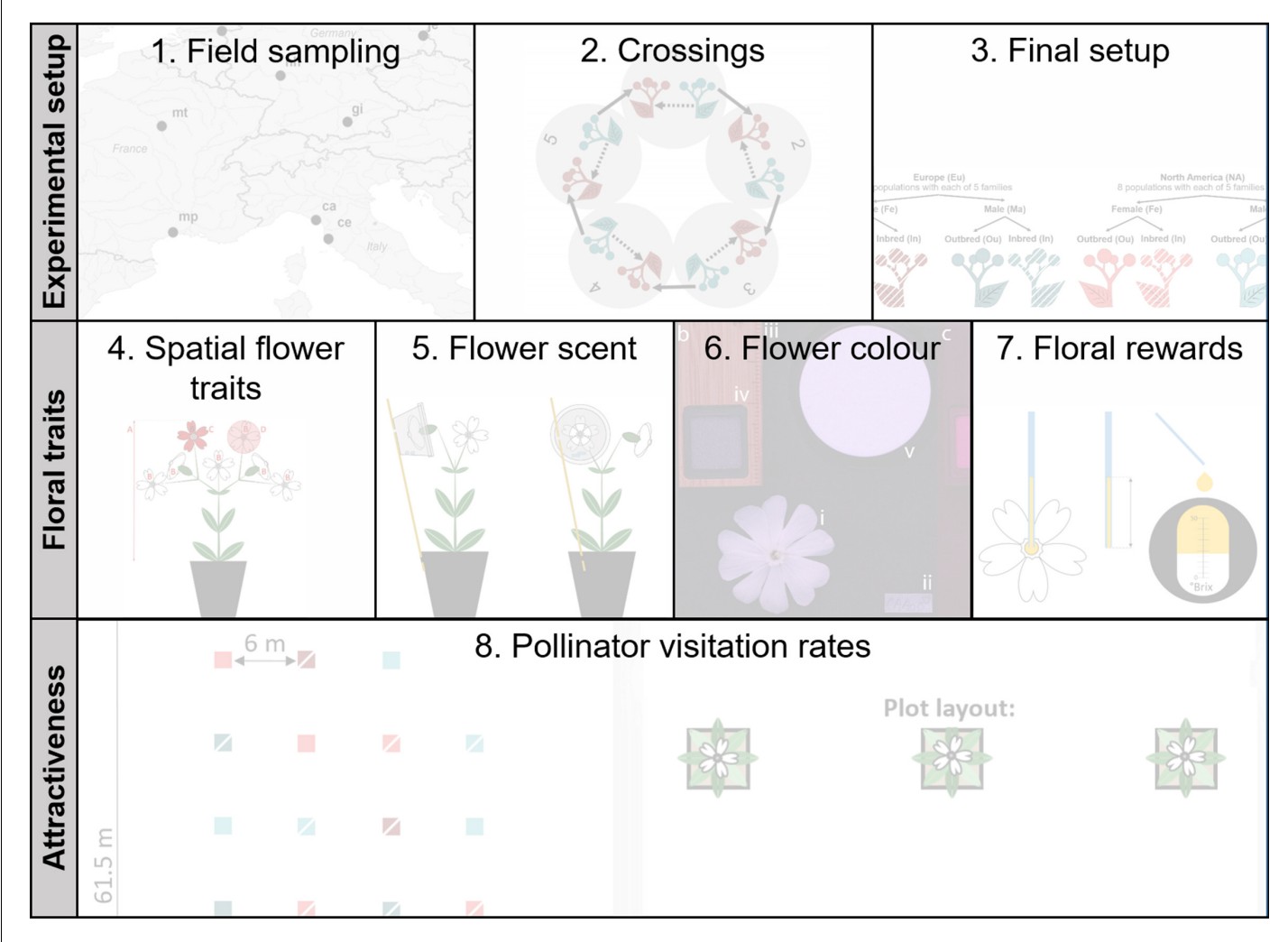

**Figure 1.** Graphical sketch of the applied methods. Each of the eight listed methodologies is illustrated in detail in a figure supplement.
The online version of this article includes the following figure supplement(s) for figure 1:

**Figure supplement 1.** Map of the geographic locations of the sampled European (left) and North American (right) *Silene latifolia* populations.
**Figure supplement 2.** Overview of the experimental crossings within each of the 16 *Silene latifolia* populations.
**Figure supplement 3.** Experimental plants.
**Figure supplement 4.** Spatial flower traits.
**Figure supplement 5.** Setup for the collection of headspace volatile organic compounds (VOC) from flowers of *Silene latifolia* (left: side view, right: front view).
**Figure supplement 6.** Setup for the acquisition of digital images for flower colour analyses.
**Figure supplement 7.** Nectar analyses.
**Figure supplement 8.** Experimental setup for pollinator observations.

Various diurnal generalist pollinators as well as crepuscular moths visit *S. latifolia* flowers. The latter, including the specialist *Hadena bicruris* Hufn. (Lepidoptera: Noctuidae), were shown to be the most efficient pollinators for *S. latifolia* (**Young, 2002**), which is the reason why we exclusively focus on nocturnal pollination in our study. All nocturnal pollinators are rewarded with nectar, while the specialist *H. bicruris* is additionally rewarded with oviposition sites. *S. latifolia* and *H. bicruris* form a well-studied nursery pollination system, in which female moths pollinate female plants while ovipositing on the flower ovaries to provide their larvae with developing seeds. Pollination services provided by male *H. bicruris* likely over-compensate the costs of seed predation by their offspring (**Labouche and Bernasconi, 2010**). A substantial fraction of floral VOC produced by *S. latifolia*

**Table 1.** Overview of locations, times, and sample sizes for data acquisition.

| Trait category | Location | Acquisition time (year, month, duration) | Nesting and intended total sample size | Realised total sample size | Replicates per group (breeding treatment × sex × origin combination) | Reason for sample size reduction |
|---|---|---|---|---|---|---|
| Spatial flower traits (synflorescence height, flower number) | Greenhouse and common garden | 2019 Jun., Jul., Aug. 5 days, respectively | Two breeding treatments × 2 sexes × 5 maternal families × 8 populations × 2 origins=320 | 316 | 36–40 | Four individuals died |
| Flower scent | Greenhouse | 2019 Jul. 8 hr | Two breeding treatments × 2 sexes × 3 maternal families × 8 populations × 2 origins=192 | 192 | 23–35 | - |
| Flower colour | Common garden | 2019 Aug. 2 weeks | Two breeding treatments × 2 sexes × 5 maternal families × 8 populations × 2 origins=320 | 286 | 23–25 | Four individuals died, no flowers available for remaining plants |
| Spatial flower traits (petal limb area and expansion) | Common garden | 2019 Aug. 2 weeks | Two breeding treatments × 2 sexes × 5 maternal families × 8 populations × 2 origins=320 | 286 | 23–35 | Four individuals died, no flowers available for remaining plants |
| Pollinator visitation rates | Field site | 2020 May–Jul. 8 weeks | Two breeding treatments × 2 sexes × 5 maternal families × 8 populations × 2 origins=320 | 316 | 36–40 | Four individuals died |
| Floral rewards | Common garden | 2020 Aug. 4 weeks | Two breeding treatments × 2 sexes × 5 maternal families × 8 populations × 2 origins=320 | 280 | 30–40 | Four individuals died, no flowers available for remaining plants |

triggers antennal and behavioural responses in male and female *H. bicruris* moths (*Dötterl et al., 2006*). The activity of *H. bicruris* peaks at dusk between May and July (*Bopp and Gottsberger, 2004*). *H. bicruris* is abundant in 90% of European *S. latifolia* populations but has not yet been introduced to North America. Other nocturnal moths including the specialist *Hadena ectypa* Morrison (Lepidoptera: Noctuidae) provide main pollination services to *S. latifolia* in the invaded range without imposing costs by seed predation (*Young, 2002*; *Castillo et al., 2014*).

## Plant material

We collected seed capsules from five female individuals (maternal families) in each of eight European and eight invasive North American *S. latifolia* populations (*Figure 1*; *Figure 1—figure supplement 1*). Seeds from all maternal families (consisting of full-sibs and/or half-sibs, hereinafter referred to as sibs) were germinated and plants were grown under controlled greenhouse conditions for experimental crossings within populations. Each female individual from the P-generation received pollen from a male derived from the same maternal family (inbreeding) and pollen from a male derived from a different maternal family within the same population (outcrossing) at separate flowers (*Figure 1—figure supplement 2*). During the crossings, plants were kept at randomised positions in the greenhouse. Female flower buds were covered with mesh bags prior to opening until fruit maturation and opened flowers were released from bags only for directed pollen transfer. The field sampling, rearing conditions, and experimental crossing are described in detail in *Schrieber et al., 2019a*, *Schrieber et al., 2019b*. Seeds were dried and stored at room temperature until further use.

For the experiment, we grew plants from the F1 generation under greenhouse conditions (16/8 hr light/dark at 20/10°C±6°C). After the onset of flowering, we randomly chose one female and one male individual per breeding treatment (inbred, outbred) × maternal family (1−5)×population (1−8) × origin (Europe, North America) combination, resulting in 320 plant individuals for the experiment (*Figure 1—figure supplement 3*). Using these individuals, we assessed the combined effects of breeding treatment, plant sex, and population origin on different flower traits and pollinator visitation rates over the summers 2019 and 2020. Plants were grown in 3 L (2019) and 6 L (2020) pots filled with a 3:1 mixture of potting soil (TKS2 Instant Plus, Floragrad, Oldenburg, Germany) and pine bark (Pine Bark 1–7 mm, Neede, Oosterbeek Humus Producten, The Netherlands). They were kept in pots with randomised positions either in the greenhouse, a common garden in Kiel, Germany (Europe) with sealed ground (54.346794°N, 10.107990°E, 19 m elevation) or a field site in Kiel, Germany (Europe), covered by an extensively used meadow (54.347742°N, 10.107661°E, 19 m elevation) for different parts of data acquisition. For an overview of the time schedule, locations, and exact sample sizes for data acquisition, see *Table 1*. Plants received water and fertilisation (UniversolGelb 12-30-12, Everris-Headquarters,Geldermalsen, The Netherlands) when necessary for the entire experimental period and were prophylactically treated with biological pest control agents under greenhouse conditions to prevent thrips (agents *Amblyseius barkeri* and *Amblyseius cucumeris*) and aphid (agent *Chrysoperla carnea*) infestation (Katz Biotech GmbH, Baruth, Germany).

## Floral traits

### Spatial flower traits

We determined the maximum height of synflorescences above ground level and the number of fully opened flowers per individual (*Figure 1—figure supplement 4a*). These traits were acquired thrice, in June, July, and August 2019 to account for phenological variation. For statistical analyses, these data were averaged over the three time points at the individual level. The size of *S. latifolia* flowers was not assessed via the length of their petal limbs as in previous studies, since this estimate does not account for the severe variation in their overall shape (*Figure 1—figure supplement 4b*). Instead, we assessed the exact area covered by all petal limbs and the expansion of the corolla (i.e., the area covered by the smallest possible circle drawn around all five petal limbs) (*Figure 1—figure supplement 4a*). Both traits were derived from digital images taken from one well-developed and fully opened flower per plant (see Flower colour section for further details) using the software ImageJ 1.47 t (*Rueden et al., 2017*).

## Flower scent

For characterisation of flower scent, we trapped the headspace VOC of *S. latifolia* flowers on absorbent polydimethylsiloxane (PDMS) tubing following the method of *Kallenbach et al., 2014*, *Kallenbach et al., 2015*. We placed the plants in a spatial distance of 50 cm to one another in the greenhouse and maintained high air ventilation 1 week prior to and during VOC collection. We selected one well-developed flower per individual and enclosed it in a VOC collection unit (*Figure 1—figure supplement 5*). The collection units consisted of polypropylene cups with lids (50 mL, Premium Line, Offenburg, Tedeco-Gizeh, Germany), both having holes (diameter 15 mm) to prevent heat and waterlogging. They were fixed via wooden sticks at the exterior of the plant pot. In addition, 14 control collection units were fixed on empty plant pots and positioned throughout the greenhouse. Prior use, the absorbent PDMS tubes (length 5 mm, external diameter 1.8 mm, internal diameter 1 mm; Carl Roth, Karlsruhe, Germany) were cleaned with solvents and heat as described in *Kallenbach et al., 2014*. Two PDMS tubes were added to each collection unit and remained in the floral headspace between 9 p.m. and 5 a.m., which is the time of peak scent emission in *S. latifolia* (*Dötterl and Jürgens, 2005*). Afterwards, the PDMS tubes were removed and stored at −20°C in sealed glass vials until analysis via thermal desorption–gas chromatography–mass spectrometry (TD-GC-MS, TD 30 – GC 2010plus – MS QP2020, Shimadzu, Kyoto, Japan).

All samples were measured in a single trial in a fully randomised order. Trapped VOC were desorbed from PDMS tubes for 8 min at 230°C under a helium flow of 60 mL min$^{-1}$ and adsorbed on a Tenax cryo-trap with a temperature of –20°C. From the trap, compounds were desorbed at 250°C for 3 min, injected to the GC in a 3:10 split mode, and migrated with a helium flow of 1.6 mL min$^{-1}$ on a VF5-MS column (30 m × 0.25 mm + 10 m guard column, Agilent Technologies, Santa Clara, CA). The GC temperature program started at 40°C for 5 min and increased to 125°C at a rate of 10°C min$^{-1}$ with a hold time of 5 min and to 280°C at a rate of 30 °C min$^{-1}$ with a hold time of 1 min. Line spectra (30–400 m/z) of separated compounds were acquired in quadrupole MS mode. An alkane standard mix (C8-C20, Sigma-Aldrich,Darmstadt, Germany) was analysed under the same conditions in order to calculate Kovats retention indices (KI) for targeted compounds (*Kováts, 1958*).

Compounds were identified by comparing the KI and mass spectra with those of synthetic reference compoundsor with library entries of the National Institute of Standards and Technology (NIST) (*Smith et al., 2004*), Pherobase (*El-Sayed, 2011*), the PubChem database (*Kim et al., 2016*), and *Adams, 2007*. Control samples (collection units without flowers), and blanks (cleaned PDMS tubes) were used to identify and exclude contaminations, leaving a total number of 70 VOC (*Supplementary file 1*). Compounds were not quantified but the intensity of the total ion chromatogram of peaks was compared among treatment groups (hereinafter referred to as intensity). A linear relationship among peak areas and compound concentrations has been validated for the passive sorption method in *Kallenbach et al., 2014*. The intensities of VOC were not corrected for flower size because we wanted to capture all variation in scent emission that is relevant for the receiver, that is, the pollinator. For targeted statistical analyses, we focused on those VOC that evidently mediate communication with *H. bicruris* according to *Dötterl et al., 2006*. We analysed the Shannon diversity per plant (calculated with R-package: vegan v.2.5–5, *Oksanen et al., 2019*) for 20 floral VOC in our data set that were shown to elicit electrophysiological responses in the antennae of *H. bicruris* (*Supplementary file 1*). Moreover, we analysed the intensities of three lilac aldehyde isomers, which trigger oriented flight and landing behaviour in both male and female *H. bicruris* most efficiently when compared to other VOC in the floral scent of *S. latifolia*. Furthermore, *H. bicruris* is able to detect the slightest differences in the concentration of these three compounds at very low dosages (*Dötterl et al., 2006*).

## Flower colour

Flower colour was quantified using a digital image transformation approach that accounts for the visual system of the pollinator as well as natural light conditions (*Troscianko and Stevens, 2015*). Images were acquired in the common garden after plants had acclimated to ambient light conditions for 3 weeks. All images were taken during 1 hr of dusk time on rain-free days in order to fit the natural light conditions perceived by *H. bicruris* (*Bopp and Gottsberger, 2004*). We picked one well-developed, fully opened flower per plant and inserted it into a black ethylene vinyl acetate platform

equipped with two reflectance standards (PTFB 10%; Spectralon 99% Labsphere, Congleton, UK) and a size standard (*Figure 1—figure supplement 6*). The platform had a fixed location in the field and was oriented towards the setting sun. Raw images were taken with a digital camera (Samsung NX1000,Suwon, South Korea, ) converted to full spectrum sensitivity (300–1000 nm) via removal of the sensor's filter and fitted with an ultraviolet (UV) sensitive lens (Nikon EL 80-mm, Japan). We took images in the visible and in the UV part of the light spectrum by fitting an UV and infrared (IR) blocking filter (UV/IR Cut, transmittance 400–700 nm, Baader Planetarium,Reutlingen, Germany) and an UV pass plus IR block filter (U-filter, transmittance 300–400 nm, Baader Planetraium,Reutlingen, Germany) to the lens, respectively. All images were taken as RAWs with an aperture of 5.6, an iso of 800, and a shutter speed varying according to light conditions.

Images were processed using the Multispectral Image Calibration and Analysis (MICA)-Toolbox plugin (*Troscianko and Stevens, 2015*) in ImageJ 1.47 t (*Rueden et al., 2017*). They were linearised to correct for the non-linear response of the camera to light intensity and equalised with respect to the two light standards in order to account for variation in natural light perceived among images (*Stevens et al., 2007*). All petals were selected for analysis, and the reproductive organs and paracorolla were omitted. Linearised images were then mapped to the visual system of a nocturnal moth. As the visual system of *H. bicruris* is unexplored, we used the tri-chromatic visual system of *Deilephila elpenor* L. (Lepidoptera: Sphingidae), which includes three rhodopsins with absorption maxima of 350 nm (UV), 440 nm (blue), and 525 nm (green) (*Johnsen et al., 2006*). We considered this system to be comparable to that of *H. bicruris*, given the similar activity behaviour of adults, morphological similarity of the preferred plant species (*Lonicera periclymenum* L. [Caprifoliaceae] with white-creamy funnel-shaped flowers) and overlapping distribution ranges. We fitted the images to a cone catch model incorporating (i) the spectral sensitivity of our Samsung NX1000-Nikkor EL 80 mm 300–700 nm camera (data derived from *Troscianko and Stevens, 2015*); (ii) the spectral sensitivities of the three photoreceptors in the *D. elpenor* compound eye (data derived from *Johnsen et al., 2006*); and (iii) the spectral composition of sun light during dusk (data derived from *Johnsen et al., 2006*).

## Floral rewards

As moths forage on liquids only, we measured nectar as floral reward (*Figure 1—figure supplement 7*). We selected one well-developed, closed flower bud per plant in the common garden and enclosed it in a transparent mesh bag (Organza mesh bags, Saketos, Sieniawka, Poland) until harvest to avoid pollination and nectar removal. All flowers were harvested at noon of the fourth day after opening and were stored immediately at 4°C until processing to prevent further nectar secretion. Nectar was extracted into 1–2 μL microcapillary tubes (Minicaps NA-HEP, Hirschmann Laborgeräte, Eberstadt, Germany). The length of the nectar column was measured with a calliper to determine the exact volume. Nectar sugar content was analysed with a refractometer adjusted for small sample sizes (Eclipse Low Volume 0–50°brix, Bellingham and Stanley, UK). Since nectar volume trades off against nectar quality in pollinator attraction (*Cnaani et al., 2006*), we addressed floral rewards in *S. latifolia* via the total amount of sugar excreted per flower as calculated based on the following equation: $\text{gsugar} = \text{volume[L]} * \left(°\text{brix} * \left(\frac{1+4.25*°\text{brix}}{1000}\right) * 10\right)$.

## Pollinator visitation rates

We quantified visits by crepuscular pollinators belonging to the order of Lepidoptera at the field site. For this purpose, plants were arranged in plots (1.5 m × 1.5 m, distance among plants = 0.5 m) that consisted of eight individuals representing all populations from one breeding treatment × sex × origin combination. Each of the possible combinations (N = 8) was replicated five times at the level of maternal families, resulting in a total number of 40 plots (N = 320 plants in total). Plots were spaced from each other at a distance of 6 m in order to provide pollinators with the choice of visiting specific breeding treatment × sex × origin combinations (*Glenny et al., 2018*). The position of plots and plants within plots was fully randomised (*Figure 1—figure supplement 8*). We performed 14 observation trials between May and July to cover the annual peak activity of *H. bicruris* (*Bopp and Gottsberger, 2004*). Each trial comprised 5 min observation time for each of the plots (total observation time: 2800 min, observation time per plot: 70 min) and was completed within 1 hr in the dawn time by four observers. The exact daytime of observation was acquired at the plot level for each of

the trials. Plant and flower visits were determined at the plant individual level. If a moth had first contact with a flower, this was counted as a plant visit. The number of approached flowers per plant during a visit was counted until a moth either left or switched to another plant. The number of plant and flower visits per trial was averaged at the plot level for further analyses. The number of visiting moth individuals and moth species was not determined. The vast majority of visits were performed by *H. bicruris* (personal observation).

Please note that North American *S. latifolia* populations were tested in their 'away' habitat only and that the observed plant performance and pollinator visitation rates can thus provide no direct implications for their 'home' habitat. However, we neither aimed at elaborating on the invasion success of *S. latifolia* nor on adaptive differentiation among European and North American populations, but at investigating inbreeding effects on plant-pollinator interactions in multiple plant populations in a common environment. Given the close taxonomic relationship of *H. bicruris* (main pollinator in Europe) and *H. ectypa* (main pollinator in North America) (*Young, 2002*; *Castillo et al., 2014*), the behavioural responses of the former species to variation in the quality of its host plant were considered to overlap sufficiently with responses of the latter species.

## Statistical analyses

All statistical analyses were performed in R v4.0.3 (*R Development Core Team, 2020*) with (generalised) linear mixed effects models (LMMs: R-package lme4 v1.1–23, *Bates et al., 2014*, GLMMs: R-package glmmTMB v1.0.2.1, *Brooks et al., 2017*). Models for responses reflecting spatial flower traits, floral scent, colour, and rewards included the predictors breeding treatment, sex, and origin, as well as all possible interactions among these factors. The latitudinal coordinate of the population origin was included as covariate in all models, whereas the exact age of the plant individuals (accounts for difference of 12 days in planting date) was included only in models for flower scent, which was acquired in early phases of the experiment. Both covariates were centred and scaled (i.e., subtraction of mean and division by standard deviation). The random effects for floral trait models were population, affiliation of paternal plant in P-generation to field collected family nested within population, and affiliation of maternal plant in P-generation to field collected family nested within population. Models for pollinator visitation rates included the predictors breeding treatment, sex, and origin, as well as all possible interactions among them, the covariate daytime (centred and scaled), and the random effects of plot and trail (latitude of population origin, population, maternal and paternal affiliation not included, since data were averaged on plot level, see Pollinator visitation rates section). Several of the described models included count data responses with an access of zeroes (intensities of lilac aldehydes and pollinator visitation rates). These models were additionally fitted with zero inflation formulas. The fit of lilac aldehydes models was best when including only an intercept model for zero inflation, whereas the fit of pollinator visitation rate models was best when including the same predictors and random effects in the conditional and zero inflation part of the model.

All of the described models (*Table 1*) were validated based on checking plots (quantile-quantile, residual versus fitted) and tests provided in the R-package DHARMa v0.3.3.0 (*Hartig, 2020*). Sum-to-zero contrasts were set on all factors for the calculation of type III ANOVA tables based on Wald $\chi^2$ tests (R-package: car v3.0–10, *Fox and Weisberg, 2018*). If origin, breeding treatment and/or sex were involved in significant interactions, we calculated post hoc contrasts on the estimated marginal means of their levels within levels of other factors involved in the respective interaction (R-package: emmeans v1.5.1, *Lenth, 2020*). Variance components were extracted from all models using the R-package insight (*Lüdecke et al., 2019*) and are summarised in *Figure 2—figure supplement 1*. Multivariate statistical analyses of the full VOC dataset are summarised in *Figure 2—figure supplement 2*.

## Results

### Floral traits

Spatial flower traits of *S. latifolia* varied pronouncedly between plants of different breeding treatments, sexes, and population origins (*Table 2*). Synflorescences of inbreds had lower maximal height above ground than those of outbreds (p<0.001, $\chi^2_{(1DF)}$=37.31, *Figure 2a*). Flower number

(*Figure 2b*) was higher in plants from North America than Europe (p=0.005, $\chi^2_{(1DF)}$=8.01) and additionally depended on the interaction breeding treatment × sex (p=0.003, $\chi^2_{(1DF)}$=8.99). Inbred plants generally produced fewer flowers than outbreds, and this effect was more severe in females (35% reduced by inbreeding, $p_{post}$ <0.001) than males (12% reduced by inbreeding, $p_{post}$ = 0.011). The number of flowers produced was lower in male than female plants in both inbreds (78% reduced in females, $p_{post}$ <0.001) and outbreds (71% reduced in females, $p_{post}$ <0.001). The area of petal limbs (*Figure 2c*) was smaller in female than male plants (p<0.001, $\chi^2_{(1DF)}$=51.35) and reduced by inbreeding (p=0.002, $\chi^2_{(1DF)}$=9.25). The expansion of the corolla depended on the interaction breeding treatment × sex (p=0.004, $\chi^2_{(1DF)}$=8.17). Inbreeding reduced corolla expansion in females by 17% ($p_{post}$ <0.001) but had no effect in male plants, and differences between sexes in corolla expansion were consequently apparent in inbreds (23% lower in females than males, $p_{post}$ <0.001) but not in outbreds (*Figure 2d*). Corolla expansion additionally depended on the interaction sex × origin (p=0.009, $\chi^2_{(1DF)}$=6.86). It was lower in female than male plants in populations originating from North America only (23% lower in females, $p_{post}$ <0.001).

Breeding treatment, sex, and population origin affected floral VOC in *S. latifolia* interactively (*Table 2*). The Shannon diversity of those VOC known to elicit antennal responses in *H. bicruris* depended on the interaction breeding treatment × origin (p=0.016, $\chi^2_{(1DF)}$=5.83, *Figure 2e*). Inbreeding reduced the Shannon diversity of these VOC by 7% in European plants ($p_{post}$ = 0.013) but had no significant effect on the Shannon diversity of the VOC in plants from North America. The intensity of lilac aldehyde A depended on the interaction breeding treatment × sex × origin in the conditional model (p=0.025, $\chi^2_{(1DF)}$=5.03, *Figure 2f*). Post hoc comparisons yielded a marginally significant lower intensity of this compound in inbred than outbred females in plants from North America (41% reduced by inbreeding, $p_{post}$ = 0.056) but no further differences occurred among other groups. Similar non-significant trends were observed for the other lilac aldehyde isomers (*Supplementary file 1*). Multivariate statistical analyses of 20 *H. bicruris* active VOC and all 70 VOC detected in *S. latifolia* revealed no clear separation of floral headspace VOC patterns for any of the treatments (*Figure 2—figure supplement 2*). In summary, the combined effects of breeding treatment, sex, and range on floral scent were rather weak.

The proportion of flower colour detectable for crepuscular moths and the sugar excreted as reward with nectar were independent of breeding treatment and population origin but exhibited differences between plants of different sex (*Table 2*). Male flowers reflected more light in the spectrum detectable by the UV receptor (350 nm) (p<0.001, $\chi^2_{(1DF)}$=41.92, *Figure 2g*) and the blue receptor (440 nm) (p<0.001, $\chi^2_{(1DF)}$=39.59, *Figure 2h*) of moths than flowers of females. Likewise, the amount of sugar excreted with nectar was higher in male than female plants (p<0.001, $\chi^2_{(1DF)}$=14.16, *Figure 2i*).

## Pollinator visitation rates

The number of pollinator visits per plant by moths was shaped by the interaction breeding treatment × sex × origin in the conditional model (p=0.016, $\chi^2_{(1DF)}$=5.84, *Table 2*, *Figure 3a*). Post hoc comparisons yielded that plant visits were reduced by 79% following inbreeding in female plants from North America ($p_{post}$ = 0.007), but unaffected by inbreeding in European females and males from both origins. Moreover, plant visits were fewer in female than male plants in European outbreds (83% fewer in females, $p_{post}$ < 0.001), European inbreds (78% fewer in females, $p_{post}$ = 0.001), and North American inbreds (87% fewer in females, $p_{post}$ < 0.001) as well as 77% lower in plants from Europe than North America in outbred females ($p_{post}$ = 0.014). The number of flowers approached per plant visit was likewise shaped by the interaction breeding treatment × sex × origin in the conditional model (p=0.001, $\chi^2_{(1DF)}$=10.61, *Figure 3b*, *Table 2*). Post hoc comparisons yielded that flower visits were 88% lower for inbred than outbred females ($p_{post}$ = 0.001) but 64% higher for inbred than outbred males ($p_{post}$ = 0.031) in plant populations from North America, whereas flower visits were unaffected by inbreeding in European male and female plants. Moreover, flower visits were reduced in females relative to males for European outbred plants (83% reduction in females, $p_{post}$ = 0.003), European inbreds (73% reduction in females, $p_{post}$ = 0.027), and North American inbreds (90% reduced in females, $p_{post}$ = 0.002) but 59% higher in females than males in outbreds from North America ($p_{post}$ < 0.001). Finally, flower visits were higher in North American than European outbred female plants ($p_{post}$ = 0.001) but lower in European than North American outbred males ($p_{post}$ = 0.001). Both the number of plant and flower visits depended on the interaction of breeding

**Table 2.** Overview and results of statistical analyses with (generalised) linear mixed effects models.

The table summarises the model types and error distributions used for each of the responses (printed in subscript), the parameter estimates on the link function scale with significance levels assessed based on Wald $\chi^2$ tests for all fixed effects (***p<0.001, **p<0.01, and *p<0.05 printed in bold), and random effect variances (printed in italic). For zero inflated responses, estimates from the conditional model parts appear in the first line and estimates from zero inflation model parts in the second line. All listed fixed effects consume 1 degree of freedom.

| | Intercept | Btmt [outbred – inbred] | Sex [female – male] | Origin [Europe – US] | Btmt × sex | Btmt × origin | Sex × origin | Btmt × sex × origin | Latitude | Plant age | Obs. time | Pop | Pop:mother | Pop: father | Plot | Obs. trial |
|---|---|---|---|---|---|---|---|---|---|---|---|---|---|---|---|---|
| *Spatial flower traits* | | | | | | | | | | | | | | | | |
| Synflorescence height$_{LMM(G)}$ | 81.03 | **3.56*** | –0.40 NS | 1.84 | 0.11 | 0.67 | 1.07 NS | –0.15 | 1.00 | Nt. | Nt. | *5.42* | *16.76* | *0.00* | Nt. | Nt. |
| No. flowers$_{GLMM(NBQ)}$ | 2.61 | **0.14*** | **–0.70*** | **–0.25** | –0.04 | **0.08** | 0.01 | 0.01 | 0.12 | Nt. | Nt. | *0.03* | *0.02* | *0.02* | Nt. | Nt. |
| Petal limb area$_{MM(G)}$ | 2.76 | **0.16** | **–0.38*** | 0.07 | 0.04 | 0.06 | 0.10 | 0.03 | 0.19 | Nt. | Nt. | *0.15* | *0.08* | *0.00* | Nt. | Nt. |
| Corolla expansion$_{LMM(G)}$ | 5.21 | **0.21** | **–0.44*** | 0.33 | 0.02 | **0.23** | **0.21** | 0.11 | 0.06 | Nt. | Nt. | *0.29* | *0.30* | *0.00* | Nt. | Nt. |
| *Flower scent traits* | | | | | | | | | | | | | | | | |
| Shannon index VOC$_{LMM(G)}$ | 1.86 | 0.03 | **–0.13*** | –0.06 | 0.04 | 0.01 | 0.02 | 0.01 | 0.03 | **–0.02** | Nt. | *0.01* | *0.00* | *0.00* | Nt. | Nt. |
| Lilac aldehyde A$_{ZI-GLMM(NBQ)}$ | 15.11 | 0.02 | –0.06 | 0.02 | –0.08 | 0.01 | –0.04 | **–0.16** | 0.01 | –0.08 | Nt. | *0.01* | *0.00* | *0.00* | Nt. | Nt. |
| | –1.93*** | Nt. | Nt. | Nt. | Nt. | Nt. | Nt. | Nt. | Nt. | Nt. | Nt. | *Nt.* | *Nt.* | *Nt.* | Nt. | Nt. |
| Lilac aldehyde B/C$_{ZI-GLMM (NBL)}$ | 15.78 | –0.08 | –0.09 | 0.12 | –0.10 | 0.06 | –0.01 | –0.03 | –0.09 | –0.03 | Nt. | *0.01* | *0.03* | *0.00* | Nt. | Nt. |
| | –1.93*** | Nt. | Nt. | Nt. | Nt. | Nt. | Nt. | Nt. | Nt. | Nt. | Nt. | *Nt.* | *Nt.* | *Nt.* | Nt. | Nt. |
| Lilac aldehyde D$_{ZI-GLMM(NBQ)}$ | 13.70 | –0.05 | 0.14 | –0.07 | –0.16 | –0.09 | –0.06 | –0.08 | 0.08 | –0.17 | Nt. | *0.00* | *2.75* | *0.47* | Nt. | Nt. |
| | 0.02 NS | Nt. | Nt. | Nt. | Nt. | Nt. | Nt. | Nt. | Nt. | Nt. | Nt. | *Nt.* | *Nt.* | *Nt.* | Nt. | Nt. |
| *Flower colour* | | | | | | | | | | | | | | | | |
| Reflectance UV$_{LMM(G)}$ | 2.04 | –0.05 | **–0.27*** | 0.05 | 0.08 NS | 0.04 | –0.04 | 0.02 | 0.01 | Nt. | Nt. | *0.01* | *0.03* | *0.01* | Nt. | Nt. |
| Reflectance blue$_{LMM(G)}$ | 26.72 | –0.02 | **–0.95*** | 0.10 | 0.05 | 0.24 | 0.21 | 0.14 NS | 0.47 | Nt. | Nt. | *0.47* | *0.85* | *0.00* | Nt. | Nt. |
| Reflectance green$_{LMM(G)}$ | 39.87 | –0.10 | –0.19 | 0.40 | 0.17 | 0.26 | 0.12 | 0.31 NS | 0.05 | Nt. | Nt. | *1.12* | *0.37* | *0.07* | Nt. | Nt. |
| *Floral rewards* | | | | | | | | | | | | | | | | |
| Excreted sugar$_{LMM(G)}$ | 5.70 | 0.03 | **–0.18*** | –0.08 | –0.02 | 0.04 | –0.02 | 0.04 | 0.04 | Nt. | Nt. | *0.07* | *0.00* | *0.00* | Nt. | Nt. |
| *Pollinator visitation* | | | | | | | | | | | | | | | | |
| No. plant visits$_{ZI-GLMM(P)}$ | 0.28 | 0.09 | **–0.67*** | 0.09 | –0.20 | 0.22 | –0.16 | **–0.29** | Nt. | Nt. | **–0.11** | *Nt.* | *Nt.* | *Nt.* | *0.27* | *0.15* |
| | –0.48 NS | 0.05 | 0.29 | –0.21 | –0.11 | –0.11 | 0.10 | **–0.46** | Nt. | Nt. | 0.10 | *Nt.* | *Nt.* | *Nt.* | *0.14* | *0.44* |
| No. flowers visited$_{ZI-GLMM(P)}$ | 0.79 | 0.04 | **–0.56*** | –0.03 | –0.23 | **0.34** | –0.22 | **–0.46** | Nt. | Nt. | **0.19*** | *Nt.* | *Nt.* | *Nt.* | *0.68* | *0.23* |
| | –0.09 NS | –0.04 | **0.47** | –0.23 | –0.02 | –0.19 | 0.17 | **–0.31** | Nt. | Nt. | **0.47** | *Nt.* | *Nt.* | *Nt.* | *0.16* | *0.47* |

Abbreviations. Btmt: breeding treatment, LMM: linear mixed effects model, (NBQ): negative binomial distribution with quadratic parametrisation and log-link, (NBL): negative binomial distribution with linear parametrisation and log-link, No.: number, Nt.: not tested, GLMM: generalised linear mixed effects model, Obs: observation, (P): Poisson distribution with log-link, Pop: population, ZI-GLMM: zero inflation generalised mixed effects model, (G): Gaussian distribution with identity link.

treatment × sex × origin in the zero inflation part of the model as well (*Table 2*, *Figure 3—figure supplement 1*). The direction and magnitude of these effects did not contrast with the conditional models.

## Discussion

Using an integrated methodological approach, we observed that (i) inbreeding compromises several flower traits in *S. latifolia*. The magnitude of these effects depended partially on (ii) plant sex, which demonstrates that the intrinsic biological differences between males and females shape the consequences of inbreeding in dioecious plant species as they are filtered through the selective environment. Inbreeding effects also depended on (iii) origin in a way indicating that divergent evolutionary histories have shaped the underlying genetic architecture. Finally, our study showed that (iv) the effects of inbreeding, sex, and origin on pollinator visitation rates specifically mirrored variation in floral scent, which yields interesting insight into the relative importance of different floral traits in shaping the behaviour of crepuscular moths.

### Inbreeding compromises floral traits

In partial accordance with our first hypothesis, inbreeding compromised several, but not all floral traits in *S. latifolia*. Spatial flower traits suffered most strongly from inbreeding in males and females from both origins (*Figure 2a–c*). These results are in line with previous studies on hermaphroditic, self-compatible species (*Ivey and Carr, 2005*; *Glaettli and Goudet, 2006*) and support that the complex genetic architecture underlying such traits (*Feng et al., 2019*) gives rise to dominance and over-dominance effects at multiple loci. The chemodiversity and abundance of floral VOC involved in communication with *H. bicruris* moths was reduced in a sex- and origin-specific manner in inbred relative to outbred *S. latifolia* (*Figure 2e–g*), while the full floral scent profile exhibited no differences among inbreds and outbreds (*Figure 2—figure supplement 2*). So far, lower emissions of floral VOC in inbreds have been reported for only few plant species pollinated by diurnal generalists (*Ferrari et al., 2006*; *Haber et al., 2019*; *Kariyat et al., 2021*). Our study revealed such effects for plants pollinated by specialist moths that use scent as a major cue for plant location (*Riffell and Alarcón, 2013*). Dominance and over-dominance may either have directly interfered with genes involved in VOC synthesis and their regulation in *S. latifolia* or unfolded their effects by disrupting physiological homoeostasis and thereby inducing intrinsic stress that came at the cost of scent production (*Kristensen et al., 2010*; *Fox and Reed, 2011*). A recent study indicates that the effects of inbreeding on the diversity of floral VOC in our study may even have been underestimated. *Kergunteuil et al., 2021* demonstrated that porous polymers may differ in their affinity with specific VOC and hence in their sensitivity in recording variation in VOC diversity entailing blind spots. They recommend a shift in practice from the use of single to multiple porous polymers (e.g., a combination of PDMS and Poropak Q) for VOC collection in future plant ecological studies, which may uncover the full impact of plant inbreeding on the composition of floral volatiles.

In contrast to spatial flower traits and scent, flower colour and the total amount of sugar excreted with nectar exhibited no differences among inbreds and outbreds in *S. latifolia* (*Table 2*). Flower colour is a trait that has, to our knowledge, not yet been studied in the context of inbreeding, despite its crucial role in flower identification and localisation (*Garcia et al., 2019*). Our data suggest that the flower colour perceived by moths is not altered by inbreeding and generally seems to be a conserved trait in *S. latifolia*. Other species with high intraspecific variation in flower colour may be ideal models to further examine the relationship with inbreeding in the future by combining visual modelling with choice experiments (*Kelber et al., 2003*). The independence of sugar excretion from inbreeding in *S. latifolia* indicates that the strong reduction of flower number in inbreds allows compensating the quality of floral rewards via a resource allocation trade-off.

Overall, the observed inbreeding effects on floral traits were partially small and variable in their magnitude as compared to previous investigations. However, our findings highlight that even weak degrees of biparental inbreeding (i.e., one generation sib-mating) can result in an impairment of multiple flower traits that is detectable against the background of natural variation among multiple plant populations from a broad geographic region. This observation indirectly supports that the selfing syndrome (i.e., smaller, less scented flowers observed in selfing relative to outcrossing populations of hermaphroditic plant species) may not merely be a result of natural selection against

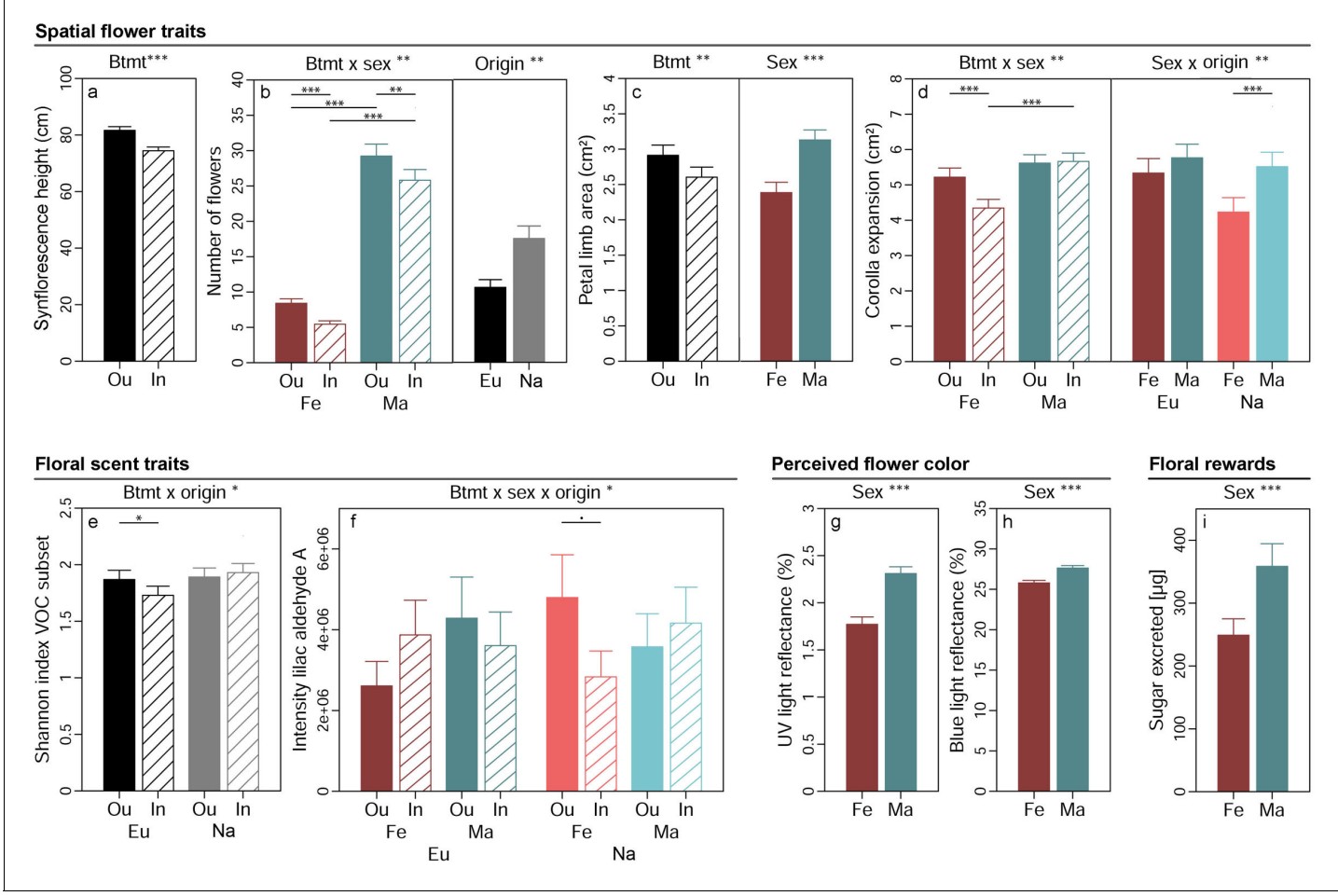

**Figure 2.** Effects of breeding treatment, sex, and origin on spatial flower traits (**a–d**), floral scent traits (**e–f**), flower colour as perceived by crepuscular moths (**g–h**), and floral rewards in *Silene latifolia*. Graphs show estimated marginal means and standard errors for outbred (Ou, filled bars) and inbred (In, open bars), female (Fe, red bars) and male (Ma, blue bars) plants from Europe (Eu, dark coloured bars) and the North America (Na, bright coloured bars). Estimates were extracted from (generalised) linear mixed effects models for significant interaction effects and main effects of factors not involved in an interaction (significance levels based on Wald $\chi^2$ tests denoted at top of plot). Interaction effect plots additionally indicate significant differences among breeding treatments, sexes, or origins within levels of other factors involved in the respective interaction (estimated based on post hoc comparisons, denoted within plots). Exact sample sizes for all traits are listed in *Table 1*. Significance levels: ***p<0.001, **p<0.01, *p<0.05, •p<0.06. The online version of this article includes the following figure supplement(s) for figure 2:

**Figure supplement 1.** Stacked bar plot illustrating the proportions of variance in floral trait responses explained by fixed effects (black) and the random effects of mother in the P-generation (dark grey), father in the P-generation (medium grey), and population (light grey), as well as the amount of unexplained variance, that is, residuals (white).

**Figure supplement 2.** Unsupervised random forest comparison MDS plots for floral headspace volatile organic compounds (VOC, left panel) and supervised random forest importance plots for mean decrease in accuracy (MDA, right panel) for all detected compounds (upper plots) and the subset for compounds that can be detected by *Hadena bicruris* (lower plots).

resource investment into floral traits, but also a direct negative consequence of inbreeding (*Andersson, 2012*). Most importantly, we observed that variation in inbreeding effects was consistent in its dependency on plant sex, which gives insight into the role of intrinsic biological differences between males and females in the expression of inbreeding depression.

## The cost of inbreeding for floral traits is higher in females than males

Males outperformed females in all floral traits, except scent production (*Figure 2*). As such, our study confirmed previously observed sexual dimorphisms in *S. latifolia* (nectar: *Gehring et al., 2004*; flower number: *Delph et al., 2010*) but also yielded contradicting results. As opposed to *Delph et al., 2010*, we observed larger instead of smaller flowers in males. This may base on the use

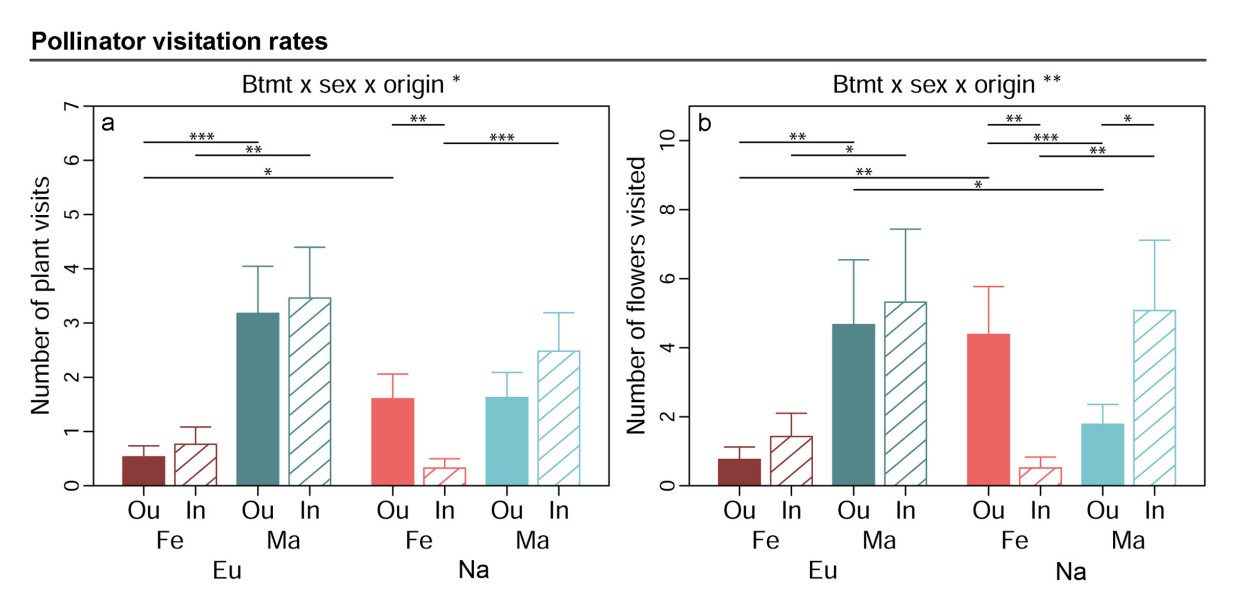

**Figure 3.** Effects of breeding treatment, sex, and origin on pollinator visitation rates in *Silene latifolia*. Graphs show estimated marginal means and standard errors for outbred (Ou, filled bars) and inbred (In, open bars), female (Fe, red bars) and male (Ma, blue bars) plants from Europe (Eu, dark coloured bars) and North America (Us, bright coloured bars). Estimates were extracted for significant interaction effects from the conditional part of generalised linear mixed effects models (significance levels based on Wald $\chi^2$-tests denoted at top of plot). Plots additionally indicate significant differences between breeding treatments, sexes, or origins within levels of other factors involved in the respective interaction (estimated based on post-hoc comparisons, denoted within plots). Exact sample sizes for all traits are listed in *Table 1*. Significance levels: ***$p<0.001$, **$p<0.01$, and *$p<0.05$.

The online version of this article includes the following figure supplement(s) for figure 3:

**Figure supplement 1.** Estimated marginal means for zero scores in pollinator visitation responses (increasing values indicate higher proportion of zeroes in response data) and standard errors for outbred (Ou, filled bars) and inbred (In, open bars), female (Fe, red bars) and male (Ma, blue bars) plants from Europe (Eu, dark coloured bars) and North America (Us, bright coloured bars).

of a size estimate that accounts for variation in flower shape or the comparably large geographic range and higher number of populations covered by our study. Moreover, we discovered a novel sexually dimorphic trait in the colour appearance of *S. latifolia* to crepuscular moths in the UV and blue light spectrum (*Figure 2g–h*). Given that moths use blue light as a major cue to start feeding on nectar (*Cutler et al., 1995*), the lower light reflectance observed for female flowers is another trait rendering them less attractive than males.

The evolution of lower female attractiveness to pollinators is driven by sex-specific resource allocation, that is, high costs for production of ovaries and seeds may restrict allocation to floral traits (*Moore and Pannell, 2011*; *Barrett and Hough, 2013*). This process may also explain the larger magnitude of inbreeding effects in female plants of *S. latifolia*, which we observed in accordance with our second hypothesis (*Figure 2b,d*). High reproductive expenditure in females may increase the frequency and intensity of resource depletion stress (e.g., drought) under field conditions (*Obeso, 2002*; *Li et al., 2004*; *Zhang et al., 2010*). Consequently, females may suffer disproportionally from inbreeding when dominance and over-dominance affect loci mediating resistance to such stress (*Fox and Reed, 2011*). The assumption that sex-specific viability selection plays a role in the expression of inbreeding depression seems likely for *S. latifolia*. Previous research elaborated that females of this species experience resource depletion stress more often than males in the late growing season during fruit maturation (*Gehring and Monson, 1994*) and that inbreeding reduces resistance to environmental stress (*Schrieber et al., 2019a*; *Schrieber et al., 2019b*). Another non-exclusive explanation for the different inbreeding effects on female versus male *S. latifolia* may be found in sexual selection. Compared to females, the reproductive success of males is more limited by the availability of mates than by the availability of resources, which results in selection for increased attractiveness to pollinators (*Moore and Pannell, 2011*; *Barrett and Hough, 2013*).

Competition for increased siring success among male plant individuals may create strong selection pressures that could rapidly purge deleterious recessive mutations in genes directly linked to attractiveness of male flowers to pollinators. Finally, a proportion of sex-specific inbreeding effects may be attributed to differential gene expression. *S. latifolia* harbours numerous genes with alleles that affect male and female fitness in opposite directions. These sexually antagonistic genes are partly subsumed in non-recombining regions of gonosomes (*Scotti and Delph, 2006*). Those located at the X-chromosome are always effectively dominant in males (XY) but may be recessive and therefore contribute to inbreeding depression in females (XX). The remaining fraction of sexually antagonistic genes is located at autosomal regions but exhibits sex-specific expression as controlled by the gonosomes (*Scotti and Delph, 2006*). These genes may exhibit systematic differences in the abundance and effect magnitudes of deleterious recessive alleles between males and females, thus contributing to sex-specific inbreeding effects.

Not only floral traits but also plant viability may exhibit sex-specific inbreeding depression in dioecious species. This could result in deviations from optimal sex ratio and, consequently, reductions of effective population sizes that accelerate local extinctions under global change (*Hultine et al., 2016*; *Rosche et al., 2018*). Future studies should aim at disentangling the relative contribution of sex-specific selection and gene expression to differences in the magnitude of inbreeding depression between males and females and at assessing their feedback on sex ratios to predict and manage these specific threats.

## Evolutionary history shapes the genetic architecture underlying inbreeding effects

Plants exhibited a general difference among geographic origins in merely one floral trait (*Figure 2b*). Indeed, we had not expected broad differences in floral traits among European and North American *S. latifolia* plants (i.e., significant main effects of origin). A sufficient overlap in the composition of pollinator communities (*H. ectypa* replaces *H. bicruris* in the invaded range, *Castillo et al., 2014*) and appropriate pre-adaptations in floral traits were probably essential for *S. latifolia* as an obligate outcrossing plant species to successfully colonise North America. As discussed in detail in previous studies, higher flower numbers in North American *S. latifolia* (*Figure 2b*) may result from changes in the selective regimes for numerous abiotic factors (*Keller et al., 2009*) or from the release of seed predation. As opposed to *H. bicruris*, *H. ectypa* pollinates North American *S. latifolia* without incurring costs for seed predation, which may result in the evolution of higher flower numbers, specifically in female plants (*Elzinga and Bernasconi, 2009*).

While adaptive differentiation among *S. latifolia* populations from different origins was not in the focus of this study, we hypothesised that North American populations purged genetic load linked to floral traits during the colonisation process (i.e., interaction breeding treatment × origin). In contrast to hypothesis iii, the magnitude of inbreeding effects was not consistently higher in European than North American populations. Instead, it was independent of origin for most floral traits, except flower scent, and either higher or lower in European plants for different scent traits (*Figure 2f–g*). These findings provide no support for recent purging events in North American populations. They rather add to evidence that the magnitude of inbreeding effects is highly specific for the traits as well as the populations or population groups under investigation (e.g., *Escobar et al., 2008*; *Angeloni et al., 2011*). This specifity roots in the composition of gene loci affected by dominance and over-dominance and is determined by the complex interplay of demographic population histories (i.e., size retractions and expansions, genetic drift, isolation, gene flow) and the selective environment (*Charlesworth and Willis, 2009*). As such, the precise mechanisms underlying variation in inbreeding effects on different scent traits across population origins of *S. latifolia* can only be explored based on comprehensive genomic resources, which are currently not available. Future studies should also incorporate field data on the abundance of specialist pollinators and extend the focus from variation in the magnitude of inbreeding effects among geographic origins to variation among populations within geographic origins and individuals within populations. This would allow a detailed quantification of geographic variation in inbreeding effects and elaborating on the causes and ecological consequences of such variation (*Thompson, 2005*; *Schrieber and Lachmuth, 2017*; *Thompson et al., 2017*).

## Inbreeding effects on floral traits cause limited feedback on pollinator visitation rates

Pollinator visitation rates partially mirrored the above-discussed variation in flower traits. They depended on the breeding treatment in a highly sex- and origin-specific manner: In North American populations, inbred females received significantly fewer plant and flower visits than outbreds, whereas flower visits were higher in inbred than outbred males (*Figure 3*). We conclude that the severe inbreeding effects on spatial flower traits alone do not necessarily reduce moth visitation rates because these effects were observed for both plant sexes and origins (*Figure 2a,b,c*). A floral trait that was negatively affected by inbreeding only in North American female plants, just like pollinator visitation rates, was the abundance of lilac aldehyde A (*Figure 2f*). The other lilac aldehyde isomers exhibited similar but non-significant trends (*Supplementary file 1*). Although these findings provide limited support for our fourth hypothesis, they yield interesting insight into the relative importance of floral traits for the behaviour of a lepidopteran specialist pollinator.

The seemingly low importance of inbreeding effects on spatial flower traits for pollinator visitation rates may be explained with the limited visual system of nocturnal insects (*van der Kooi et al., 2021*; *Sondhi et al., 2021*). Moths can likely perceive differences in the spatial arrangement of flowers on inbred versus outbred plants only when they have already approached close to them (*Barnett et al., 2018*). Our setup for pollinator observations had relatively large distances among replicate inbred and outbred plots (*Figure 1—figure supplement 8*) and hence may have not enabled a choice based on spatial flower traits by the moths. In contrast, differences in scent cues should be perceived across large distances. The antennae of *H. bicruris* can detect slight differences in lilac aldehyde concentrations at very low dosages and the compounds elicit oriented flight and landing responses in the moth more than any other VOC in the scent of *S. latifolia* flowers (*Dötterl et al., 2006*). Consequently, a low lilac aldehyde abundance may have resulted in a low attraction of moths to North American inbred female plants from the distance in our experiment. The non-significant trend for higher abundances of lilac aldehydes in inbred than outbred males from North America (*Supplementary file 1*) could also explain the corresponding variation observed in flower visitation rates. However, bioassays under more controlled conditions are needed to further evaluate a mechanistic relationship among pollinator visitation and intensities of lilac aldehydes.

In summary, our research on *S. latifolia* suggests that in addition to inbreeding disrupting interactions with herbivores by changing plant leaf chemistry (*Schrieber et al., 2019a*), it affects plant interactions with pollinators by altering flower chemistry. Our observations are in line with studies on other plant species (*Ivey and Carr, 2005*; *Kariyat et al., 2012*; *Kariyat et al., 2021*) and highlight that inbreeding has the potential to reset the equilibrium of species interactions by altering functional traits that have developed in a long history of co-evolution. These threats to antagonistic and symbiotic plant-insect interactions may mutually magnify in reducing plant individual fitness and altering the dynamics of natural plant populations under global change. As such, our study adds to a growing body of literature supporting the need to maintain or restore sufficient genetic diversity in plant populations during conservation programs.

## Acknowledgements

This study was funded by the program for the promotion of young female scientists of the Faculty of Mathematics and Natural Sciences of Kiel University. We warmly thank Ms. Koopmann, who kindly provided her private property as field site for pollinator observations. Tim Diekötter provided germination chambers and greenhouse space and Wolfgang Bilger supported us with light standards for digital image acquisition. Susanne Petersen, Stephan Doose, David Eder, Carolin Böttcher, Jorun Jess, Sarai Guadalupe Quezada-Jimenez, Verena Zajonc, Ann-Cathrin Voss, and Pia Music provided technical assistance. CM and EE acknowledge support by the research unit FOR3000, funded by the German Research Foundation (Deutsche Forschungsgemeinschaft, DFG).

# Additional information

## Funding

| Funder | Author |
|---|---|
| Kiel University, Faculty of Mathematics and Natural Sciences, program for promotion of young female scientists | Karin Schrieber |
| Kiel University, Faculty of Mathematics and Natural Sciences, program for promotion of young female scientists | Alexandra Erfmeier |

The funders had no role in study design, data collection and interpretation, or the decision to submit the work for publication.

## Author contributions

Karin Schrieber, Conceptualization, Data curation, Formal analysis, Supervision, Funding acquisition, Validation, Investigation, Visualization, Methodology, Writing - original draft, Project administration, Writing - review and editing; Sarah Catherine Paul, Resources, Supervision, Validation, Methodology, Writing - review and editing, Investigation; Levke Valena Höche, Andrea Cecilia Salas, Rabi Didszun, Jakob Mößnang, Investigation; Caroline Müller, Resources, Methodology, Writing - review and editing; Alexandra Erfmeier, Resources, Funding acquisition, Writing - review and editing; Elisabeth Johanna Eilers, Data curation; Formal analysis; Validation; Investigation; Methodology; Writing - review and editing

## Author ORCIDs

Karin Schrieber https://orcid.org/0000-0001-7181-2741
Alexandra Erfmeier https://orcid.org/0000-0002-1002-9216

## Decision letter and Author response

Decision letter https://doi.org/10.7554/eLife.65610.sa1
Author response https://doi.org/10.7554/eLife.65610.sa2

# Additional files

## Supplementary files

• Supplementary file 1. Mean ± SE abundance (e-06) of 70 volatile organic compounds (VOC) emitted by *Silene latifolia* plants in all breeding treatment × sex × origin combinations, as determined by silicone tubing headspace collection combined with thermal desorption–gas chromatography–mass spectrometry. The 20 VOC, which evidently trigger antennal responses in *Hadena bicruris* (*Dötterl et al., 2006*) and were thus analysed as a sub-dataset, are highlighted in bold.

• Transparent reporting form

## Data availability

All Data supporting this article have been deposited in Dryad. The code for all statistical analyses presented in this manuscript is deposited in Zenedo.

The following datasets were generated:

| Author(s) | Year | Dataset title | Dataset URL | Database and Identifier |
|---|---|---|---|---|
| Schrieber K, Paul SC, Höche LV, Salas AC, Didszun R, Mößnang J, Müller C, Erfmeier | 2021 | Data: Inbreeding in a dioecious plant has sex- and population origin-specific effects on its interactions with pollinators | https://doi.org/10.5061/dryad.612jm643d | Dryad Digital Repository , 10.5061/dryad.612jm643d |

| | | | | |
|---|---|---|---|---|
| A, Eilers EJ | | | | |
| Schrieber K, Paul SC, Höche LV, Salas AC, Didszun R, Mößnang J, Müller C, Erfmeier A, Eilers EJ | 2021 | Data: Inbreeding in a dioecious plant has sex- and population origin-specific effects on its interactions with pollinators | https://doi.org/10.5281/zenodo.4746164 | Zenodo , 10.5281/zenodo.4746164 |

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
