## [Decision Letter]

Our editorial process produces two outputs: (i) public reviews designed to be posted alongside the preprint for the benefit of readers; (ii) feedback on the manuscript for the authors, including requests for revisions, shown below.

Thank you for submitting your article "Sex and origin-specific inbreeding effects on flower attractiveness to specialised pollinators" for consideration by *eLife*. Your article has been reviewed by 3 peer reviewers, and the evaluation has been overseen by a Reviewing Editor and Meredith Schuman as the Senior Editor. The reviewers have opted to remain anonymous.

Essential revisions:

1. A recent study has addressed some of the questions detailed in the manuscript. So, the introduction needs to be tweaked to reflect this.

2. The authors found stronger effects of inbreeding on pollinator visitation rates in the plants from the North American (Na) origin. However, those plants were only tested in non-native habitat. As the main pollinator is also a seed predator, the Na populations could be released from the selection pressure to avoid attracting the females of this species and thus risking the loss of seeds and fitness. This issue should be addressed in the discussion explicitly and its consequences for the interpretation of the results should be considered.

3. Please revise the title according to the comments from the reviewers.

*Reviewer #1 (Recommendations for the authors):*

1. Although in a few cases inbreeding has been shown to alter single components of flower attractiveness (Ivey and Carr, 2005; Ferrari et al., 2006; Haber et al., 2019), insight into syndrome-wide effects is lacking.

This statement is no longer correct. Please refer https://bsapubs.onlinelibrary.wiley.com/doi/abs/10.1002/ajb2.1594 where the authors show flower size, pollen content, floral volatiles and floral visits are affected by inbreeding. The introduction needs to be reframed adding this and any additional notes on the novelty if needed.

2. Diurnal and nocturnal visitors and their rewards, are VOC's different during the day? There seems to be no explanation for this.

3. Line 111: This is a bit unclear; are the female floral volatiles oviposition attractants? rather than a typical floral scent?

Again, in line 189 authors discuss EAG data, but not clear whether it is from male or female moths, especially since female moths may have been using these compounds as oviposition cues.

I understand that this is not directly relevant to the current study, but it is indeed an important part of this system and should be discussed.

4. 143: What are these biological pest control agents, and for what reason they were used. There is a need to explain this since floral and foliar chemistry can be affected (differentially due to inbreeding) due to any foliar or floral herbivory. I have to say that I am surprised that such a treatment has been carried out before chemistry experiments.

5. 154: Floral volatiles are collected on a per flower basis, however, there is a huge variation for size and possibly flower mass among the treatments. The authors should explain how that would not have affected their results?

6. While the study is laser-focused on floral traits, as the authors are aware inbreeding affects the total phenotype of the plants including fitness and defense traits. For example., there are quite a few studies that have shown how inbreeding affects the plant defense phenotype. While this is irrelevant to some of the traits studied in this manuscript, it does matter for floral scent. While the floral scent is usually semi-distinct from foliar volatiles (Baldwin group, Raguso group), the volatiles also shares structural and functional similarities. The authors I think, should address how these effects might be either additive or antagonistic in nature.

7. 265: If each flower for scent extraction was of the same age (4th day) irrespective of the age of the plant (12-day variation), is it necessary to add that into the model for scent analyses in statistics?

*Reviewer #2 (Recommendations for the authors):*

Below, please find my detailed comments in the order of appearance:

Title (and throughout the manuscript): I am not sure about the term floral attractiveness. Why not "floral traits" or "pollination-linked traits" or something along these lines? You did measure floral attractiveness to pollinators, and for most traits the reduction of the floral attractive traits did not really matter for the pollinator (and Hadena seed predator?) visitation rates.

Line 24. It is unclear here both what you mean about plant attractiveness and how you mean that it may be affected by inbreeding. I think you need to establish why you think that inbreeding may play a role already here, and that you expect that traits that are involved in pollinator attraction may be negatively affected by inbreeding by citing previous work (e.g. Andersson 2012 Am J Bot).

Line 33: This sentence is a bit unclear, because you connect (Darwinian) fitness, which is a relative measure varying among individuals within a population, and inbreeding depression, which most often is used as a population measure. I wonder if instead of fitness you are focusing on e.g. population viability?

Line 43: In many cases it is unclear at what distance the floral scent is actually detected and utilized by the pollinators (cf. examples of e.g. scent nectar guides, scented nectar etc.).

Line 57-58: I think that it would be better to introduce here the plant examples where inbreeding may affect plant traits, and then make a case why it is important to study the impact of inbreeding on floral traits rather than just justifying the study by saying that it has not been done before.

Line 114: I would add a "may" between "H. biuris" and "overcompensate", because the outcome of the Silene-Hadena interaction may very likely depend on the presence (or absence) of alternative, less costly, pollinator species.

Line 135 and 269-273: Why was the factor "year" not included in these models? Would it impact the results?

Line 147. The height and number of was scored three times. Once, then again after two weeks and then again after six weeks from the first scoring? Or once, then again after two weeks and then again after six weeks after the second scoring? Please clarify.

Line 155-190. I am a bit puzzled by the floral scent collections and analyses. Usually, static collection with PMDS absorbents is used qualitatively rather than quantitatively, and in the methods it is mentioned that the compounds were not quantified but that the relative intensity of the different VOC peaks was compared. Hence, you focused on floral scent composition. This is fine, but then in the analyses of the lilac aldehydes (e.g. F), I understand it as you measure the absolute number of counts from the chromatograms.

Hence, I wonder (i) were these data stilled analysed quantitatively, and, if so, how did you validate that the number of counts was comparable across samples? The traditional method is to use dynamic headspace sampling, and then add a standard volume of a known substance to account for e.g. variable GC-MS sensitivity. Alternatively, (ii) this measure is relative (as indicated by the text and the y-axis label in the figure), but then I would like a better description of how this variable was calculated. For example, if using proportions, two samples could emit the exact same amount of the particular compound, but the proportional contribution of the same compound each overall scent bouquet could be very different, if one of the two samples include large amounts also of other scent compounds. Under such a scenario, it would be misleading to use the relative contribution of one single floral scent compound to try to predict pollinator behaviour. I apologize if I have missed something obvious here, but if so, it still makes sense to clarify how these data were obtained and what may be their limitation.

Line 335-344: Again, I was confused by the term floral attractiveness, since most traits were not that important for pollinator visitation rates. This was confusing throughout the discussion.

Line 347: Also "spatial attractiveness" is a bit misleading. Why not talk about the actual traits?

Line 445-477: I like the discussion about how the inbreeding effects on most traits, with the potential exception of floral scent, did not result in a reduced pollinator visitation rate. I think, however, that there is room for a deeper discussion here. These other traits, are they not important for pollinator attraction at all? Or were they not reduced to a large enough extent to affect pollinator visitation? Also, considering the impressive amount of data produced and the numerous predictor variables, how well can we trust that the floral scent reduction is indeed not just a spurious correlation? This is particularly relevant in the light of the comment above about whether or not we can be sure that you have detected variation in "aldehyde concentrations" (line 460). I was pondering whether there was an additional way to analyze these data by comparing effect sizes (trait variation between inbred and outcrossed plants of the same category) and ask how well these explained the variation in effect size in pollinator visitation rate between inbred and outcrossed plants of the same categories. I could not come up with a straightforward way to do so, so I realize the problem of drawing too strong conclusions based on a single data point (a slight reduction in aldehyde concentration in inbred females from North America and a corresponding reduced visitation rate on inbred North American females). I think that one important take-home message here is the lack of trait-based explanations for pollinator visitation rates (which further emphasizes the need to avoid using "floral attractiveness").

Figure S8: I wonder if there is a mistake here, because the two plots in the left panel are identical.

Finally, I wanted to reiterate the potential to study variation also at smaller scale. How consistent were the trait changes among populations from the same continent. Is it reasonable to lump these into the same category and to compare among continents. It would be very interesting to see some variance partitioning.

*Reviewer #3 (Recommendations for the authors):*

Line 2: The first sentence of the abstract is difficult to follow, consider rephrasing. In addition, while I would definitely agree that looking at effects of inbreeding on flowers in dioecious plants is most interesting and important because of the obligate outcrossing, this study does not explore whether these effects are "particularly fatal in dioecious" as there is no comparison with a hermaphroditic or monoecious plant.

I found the term spatial attractiveness (first encounter on line 8) quite confusing (later on also spatial floral traits). Wouldn't it simply be visual attractiveness? Of course this would not be an appropriate umbrella term considering that you also include number of flowers and inflorescence height.

Considering flower number only as a component of floral attractiveness is not taking full advantage of the measure. This could also be considered an important predictor of fitness and thus quite separate from the other floral morphological traits. Same could be, to some degree, said for inflorescence height, which I assume simultaneously means plant height?

In any case, I would like the authors to reconsider calling this group of floral traits "spatial".

Line 31: Change raises to increases.

Line 91: Be more specific about what you mean by spatial traits or rethink the term altogether.

Line 114: bicruris.

Line 124: Are the inbred crosses between full-sibs or half-sibs? Are seeds of one capsule always sired by the same father, and if so, were the plants from each capsule kept separate? What would be the resulting F of the inbred progeny? also correct this on line 372 if they are half-sibs.

Line 167: Check the time, does not make sense with the 8h reported in the table.

Line 285: or S8?

Line 289: Height of inflorescence was lower? This sounds like you mean the length of the corolla.

Line 293: Perhaps it would be better to report effect sizes or percent changes here. Both differences are statistically significant, and a difference in two <0.05 p-values is not really an appropriate way of reporting a difference in the effect.

Line 303: It is somewhat misleading to call this composition of floral VOCs when in reality you tested diversity and three individual compounds our of 70.

Line 404: Why would purging only affect male flowers?

Line 449: "visits were higher in inbred males." Compared to inbred females or outcrossed males?

Table 2. Ns's could be removed for clarity, a simple absence of asterisks already indicates a non-significant effect. Add degreed of freedom.

---

## [Author Response]

Essential revisions:1. A recent study has addressed some of the questions detailed in the manuscript. So, the introduction needs to be tweaked to reflect this.

We adjusted the writing in the introduction and the discussion accordingly. Introduction pp 4-5, ll 48-54:

“Although in a few cases inbreeding has been shown to alter single components of flower attractiveness (Ivey and Carr, 2005; Ferrari et al., 2006; Haber et al., 2019), insight into syndrome-wide effects is restricted to a single study. [...] It is necessary to broaden such integrated methodological approaches to other plant-pollinator systems (e.g., nocturnal specialist pollinators) and further floral traits (i.e., flower colour).”

Discussion p 19, ll 535-542:

“In summary, our research on *S. latifolia* suggests that in addition to inbreeding disrupting interactions with herbivores by changing plant leaf chemistry (Schrieber et al., 2018) it affects plant interactions with pollinators by altering flower chemistry. […] These threats to antagonistic and symbiotic plant-insect interactions may mutually magnify in reducing plant individual fitness and altering the dynamics of natural plant populations under global change.”

2. The authors found stronger effects of inbreeding on pollinator visitation rates in the plants from the North American (Na) origin. However, those plants were only tested in non-native habitat. As the main pollinator is also a seed predator, the Na populations could be released from the selection pressure to avoid attracting the females of this species and thus risking the loss of seeds and fitness. This issue should be addressed in the discussion explicitly and its consequences for the interpretation of the results should be considered.

Indeed, North American populations are tested in their “away”-habitat only and the observed plant performance and pollinator visitation rates can thus provide no direct implications for their “home”-habitat. We state this now more clearly at pp 11-12, ll 283-285. However, our design is appropriate for investigating inbreeding effects on plant-pollinator interactions in multiple plant populations in a common environment. Given the close taxonomic relationship of *H. bicruris* (main pollinator in Europe) and *H. ectypa* (main pollinator in North America), the behavioural responses of the former species to variation in the quality of its host plant was considered to overlap sufficiently with responses of the latter species as outlined at pp 11-12, ll 285-291.

The hypothesis that North American (NA) *S. latifolia* evolved higher attractiveness to female *Hadena* moths because *H. ectypa* is not able to oviposit on female plants in contrast to *H. bicruris* is indeed a highly interesting one. However, as you have outlined correctly, our study is not designed to elaborate on questions related to adaptive evolutionary differentiation among North American and European plants. Instead of addressing this hypothesis based on our data, we thus take reference to previous studies in the discussion p 17, ll 482-487:

“As discussed in detail in previous studies, higher flower numbers in North American *S. latifolia* plants (Figure 1b) may result from changes in the selective regimes for numerous abiotic factors (Keller et al., 2009) or from the release of seed predation. As opposed to *H. bicruris*, *H. ectypa* pollinates North American *S. latifolia* without incurring costs for seed predation, which may result in the evolution of higher flower numbers, specifically in female plants (Elzinga and Bernasconi, 2009).”

3. Please revise the title according to the comments from the reviewers.

The title was changed to: “Inbreeding in a dioecious plant has sex- and population origin-specific effects on its interactions with pollinators”.

Reviewer #1 (Recommendations for the authors):1. Although in a few cases inbreeding has been shown to alter single components of flower attractiveness (Ivey and Carr, 2005; Ferrari et al., 2006; Haber et al., 2019), insight into syndrome-wide effects is lacking.This statement is no longer correct. Please refer https://bsapubs.onlinelibrary.wiley.com/doi/abs/10.1002/ajb2.1594 where the authors show flower size, pollen content, floral volatiles and floral visits are affected by inbreeding. The introduction needs to be reframed adding this and any additional notes on the novelty if needed.

Thank you very much for bringing this excellent article to our attention! We adjusted the writing in the introduction and the discussion accordingly. Please consider that this article was first published at the 15^th^ of January 21, while our manuscript was submitted at the 9^th^ of January. Hence, we were not able to account for this study in the first submission. Introduction pp 4-5, ll 48-54:

“Although in a few cases inbreeding has been shown to alter single components of flower attractiveness (Ivey and Carr, 2005; Ferrari et al., 2006; Haber et al., 2019), insight into syndrome-wide effects is restricted to a single study. [...] These threats to antagonistic and symbiotic plant-insect interactions may mutually magnify in reducing plant individual fitness and altering the dynamics of natural plant populations under global change.”

2. Diurnal and nocturnal visitors and their rewards, are VOC's different during the day? There seems to be no explanation for this.

*Hadena bicruris* and other nocturnal moths are the most efficient pollinators of *S latifolia*, and the pollination syndrome of the *S. latifolia* points clearly to nocturnal pollination (p 6, ll 106-109). We therefore focussed on nocturnal pollinators and their specific rewards, which is now stated more clearly in the methods section at p 7, ll 117-120:

“Various diurnal generalist pollinators as well as crepuscular moths visit *S. latifolia* flowers. The latter, including the specialist *Hadena bicruris* Hufn. (Lepidoptera: Noctuidae), were shown to be the most efficient pollinators for *S. latifolia* (Young, 2002), which is the reason why we exclusively focus on nocturnal pollination in our study.”

To keep the focus on what has been done, we do not describe the rewards for diurnal generalists. However, we now give more detail about the rewards for nocturnal pollinators already in the study species section at p 7, ll 120-121:

“All nocturnal pollinators are rewarded with nectar, while the specialist *H. bicruris* is additionally rewarded with oviposition sites.”

Moreover, we added more information on changes in scent emission during the day at p 6, ll 107-109:

“The flowers open from dusk till mid-morning to release a scent bouquet composed of more than 60 VOC, whereby emission peaks around dusk (Dötterl et al., 2005, 2009; Mamadalieva et al., 2014). During the daytime, no measurable floral scent is emitted (Dötterl et al. 2005).”

3. Line 111: This is a bit unclear; are the female floral volatiles oviposition attractants? rather than a typical floral scent?Again, in line 189 authors discuss EAG data, but not clear whether it is from male or female moths, especially since female moths may have been using these compounds as oviposition cues.I understand that this is not directly relevant to the current study, but it is indeed an important part of this system and should be discussed.

We clarified this issue at different occasions in the methods section. Previous studies (and our study) on *S. latifolia* have shown no clear differences in the quality of floral scent between sexes. However, one study found higher total emission of VOC in males, while others found no differences. Hence, females produce no specific VOC that are used as oviposition cues but may be differentiated from males by the total amount of emitted VOC and pronounced differences in spatial flower traits. We highlight this at p 6, ll 111-116:

“*Silene latifolia* exhibits various sexual dimorphisms with male plants producing more and smaller flowers that excrete lower volumes of nectar with higher sugar concentrations as compared to females (Gehring et al., 2004; Delph et al., 2010). The quality of floral scent exhibits no clear sex-specific patterns, while male plants have been shown to emit higher or equal total amounts of VOC as compared to females in different studies (Dötterl and Jürgens 2005, Waelti et al. 2009)”.

Both male and female moths show pronounced behavioural responses to lilac aldehyde isomers and other VOC in the floral scent of *S. latifolia* (Dötterl et al., 2006). We therefore treated these VOC as typical floral scent compounds. We clarified this at p 7, ll 125-126:

“A substantial fraction of floral VOC produced by *S. latifolia* triggers antennal and behavioural responses in male and female *H. bicruris* moths (Dötterl et al., 2006).”

and p 9, ll 2010-218:

”For targeted statistical analyses, we focused on those VOC that evidently mediate communication with *H. bicruris* according to Dötterl et al. (2006). […] Furthermore, *H. bicruris* is able to detect the slightest differences in the concentration of these three compounds at very low dosages (Dötterl et al. 2006).”

4. 143: What are these biological pest control agents, and for what reason they were used. There is a need to explain this since floral and foliar chemistry can be affected (differentially due to inbreeding) due to any foliar or floral herbivory. I have to say that I am surprised that such a treatment has been carried out before chemistry experiments.

Thank you very much for highlighting this point! We used biological pest control agents in a preventive manner because *S. latifolia* is often infested by thrips and aphids under greenhouse conditions. The writing in the previous manuscript version was not clear with this regard and we changed the text at p 8, ll 157-161:

”Plants received water and fertilisation (UniversolGelb 12-30-12, Everris-Headquarters, NL) when necessary for the entire experimental period and were prophylactically treated with biological pest control agents under greenhouse conditions to prevent thrips (agent *Amblyseius barkeri* and *Amblyseius cucumeris*) and aphid (agent *Chrysoperla carnea*) infestation (Katz Biotech GmbH, GE).”

5. 154: Floral volatiles are collected on a per flower basis, however, there is a huge variation for size and possibly flower mass among the treatments. The authors should explain how that would not have affected their results?

Indeed, flower size and scent emission can be correlated. Although the question whether differences in scent emission were based on a difference in flower size is an interesting one, it seemed less relevant to us because it is unlikely that our pollinators correct their perception of a scent for the size of a flower (see also p 19, 520-526). We were rather interested in whether scent emission differs between the plant treatments and thus pollinators may chemically perceive such differences.

Moreover, we found it problematic to correct our models for flower size by including it as a covariate, which is the reason why we have not assessed this trait during scent collection. In this case, we would have corrected our scent responses for the effects of inbreeding, sex and population origin (i.e., the predictors we are interested in) because all of them determine the size of a flower (Figure 2 c,d). Hence, the inbreeding, sex and origin effects on flower scent would likely vanish. However, it is highly unlikely that the set of genes contributing to sex-, breeding treatment- and origin-based variation in flower size is exactly the same one that determines variation in scent emission per flower, which is basically the assumption underlying the model that includes flower size as a covariate. We critically mentioned the trade-off relationships and our reasoning to not correct for flower size at 9p ll 208-210:

“The intensities of VOC were not corrected for flower size because we wanted to capture all variation in scent emission that is relevant for the receiver i.e., the pollinator.”

6. While the study is laser-focused on floral traits, as the authors are aware inbreeding affects the total phenotype of the plants including fitness and defense traits. For example., there are quite a few studies that have shown how inbreeding affects the plant defense phenotype. While this is irrelevant to some of the traits studied in this manuscript, it does matter for floral scent. While the floral scent is usually semi-distinct from foliar volatiles (Baldwin group, Raguso group), the volatiles also shares structural and functional similarities. The authors I think, should address how these effects might be either additive or antagonistic in nature.

We agree that this aspect is important and therefore addressed it in further detail in the introduction at p 4 ll 34-38:

“While it is well established that inbreeding can increase a plant’s susceptibility to herbivores by diminishing morphological and chemical defences (Campbell et al., 2013; Kariyat et al., 2012; Kalske et al., 2014), its effects on plant-pollinator interactions are less well understood. Inbreeding may reduce a plant’s attractiveness to pollinating insects by compromising the complex set of floral traits involved in interspecific communication.”

Since other referees suggested to rather tone down than increase the discussion based on floral scent results, we stick to the general feedback relationship among of herbivory and pollination, rather than relating it specifically to volatiles in the discussion at p 19, ll 535-544:

“In summary, our research on *S. latifolia* suggests that in addition to inbreeding disrupting interactions with herbivores by changing plant leaf chemistry (Schrieber et al., 2018) it affects plant interactions with pollinators by altering flower chemistry. […] As such, our study adds to a growing body of literature supporting the need to maintain or restore sufficient genetic diversity in plant populations during conservation programs.”

7. 265: If each flower for scent extraction was of the same age (4th day) irrespective of the age of the plant (12-day variation), is it necessary to add that into the model for scent analyses in statistics?

Yes, this potentially confounding effect explained sufficient variation in one of our scent responses (Table 2, Shannon index) and substantially improved the model fit for lilac aldehydes (see r-script ll 562-818).

Reviewer #2 (Recommendations for the authors):Below, please find my detailed comments in the order of appearance:Title (and throughout the manuscript): I am not sure about the term floral attractiveness. Why not "floral traits" or "pollination-linked traits" or something along these lines? You did measure floral attractiveness to pollinators, and for most traits the reduction of the floral attractive traits did not really matter for the pollinator (and Hadena seed predator?) visitation rates.

We now avoid the term floral attractiveness throughout the manuscript and instead refer to “floral traits”.

Line 24. It is unclear here both what you mean about plant attractiveness and how you mean that it may be affected by inbreeding. I think you need to establish why you think that inbreeding may play a role already here, and that you expect that traits that are involved in pollinator attraction may be negatively affected by inbreeding by citing previous work (e.g. Andersson 2012 Am J Bot).

We rephrased the text at p 4, ll 24-29:

“Plant population retraction and isolation may also affect interactions with pollinators at the plant individual level by increasing inbreeding rates (Carr et al., 2014). […] Mechanistic insight into the effects of inbreeding on plant-pollinator interactions and intrinsic factors shaping the magnitude of such effects is limited but urgently required for the conservation of component species.”

Line 33: This sentence is a bit unclear, because you connect (Darwinian) fitness, which is a relative measure varying among individuals within a population, and inbreeding depression, which most often is used as a population measure. I wonder if instead of fitness you are focusing on e.g. population viability?

Indeed, we forgot to set outbred individuals as a reference, when writing about fitness declines in inbreds. We changed the sentence at p 4, ll 30-33 accordingly:

“This may enhance the phenotypic expression of deleterious recessive mutations (i.e., dominance) and reduce heterozygote advantage (i.e., over-dominance), which can result in severe declines of Darwinian fitness in inbred relative to outcrossed offspring (i.e., inbreeding depression) (Charlesworth and Willis, 2009).”

Inbreeding depression is defined and quantified as a reduction of fitness in inbred relative to outbred offspring and can be assessed on the family, population or meta-population level (Charlesworth and Willis, 2009).

Line 43: In many cases it is unclear at what distance the floral scent is actually detected and utilized by the pollinators (cf. examples of e.g. scent nectar guides, scented nectar etc.).

We removed the information in the parentheses at p 4, ll 46-48:

“These cues are particularly efficient in attracting pollinators across either long, medium or short distances (Dafni et al., 1997; Muhlemann et al., 2014) and act synergistically in determining visitation rates.”

Line 57-58: I think that it would be better to introduce here the plant examples where inbreeding may affect plant traits, and then make a case why it is important to study the impact of inbreeding on floral traits rather than just justifying the study by saying that it has not been done before.

Thank you very much for highlighting this passage! We revised the text to improve the reading flow at p 5, ll 66-70:

“If inbreeding effects on floral traits are more pronounced in female than male plants, the relative frequency of pollinator visits may be biased towards the latter sex, with devastating consequences for the effective size and persistence of populations. Studies on sex-specific inbreeding effects on floral traits are thus needed to improve the risk assessment for the conservation of dioecious plant species.”

Line 114: I would add a "may" between "H. biuris" and "overcompensate", because the outcome of the Silene-Hadena interaction may very likely depend on the presence (or absence) of alternative, less costly, pollinator species.

Done p 7, l 124.

Line 135 and 269-273: Why was the factor "year" not included in these models? Would it impact the results?

Due to the extensive number and extent of measurements, we needed 2 years to acquire the full set of considered traits. However, none of the traits was assessed in both 2019 and 2020. We clarified this at p 8, ll 149-151:

“Using these individuals, we assessed the combined effects of breeding treatment, plant sex and population origin on different flower traits and pollinator visitation rates over the summers 2019 and 2020”.

See also Table 1, p 31. Instead, some data (flower number, synflorescence height) were assessed thrice in 2019 to reduce confounding effects of differences in flowering phenology. Indeed, one possible way to analyse such repeated measure data is to include time point as a fixed factor and plant individual as a random factor. An alternative solution is averaging the data over time at the individual level and with this approach it is neither necessary nor possible to account for time point in the model. As stated at p 8, ll 164-167, we used averaged data:

“We determined the maximum height of synflorescences above ground level and the number of fully opened flowers per individual (Figure 1—figure supplement 4a). […] For statistical analyses, these data were averaged over the three time points at the individual level.”

Line 147. The height and number of was scored three times. Once, then again after two weeks and then again after six weeks from the first scoring? Or once, then again after two weeks and then again after six weeks after the second scoring? Please clarify.

It was scored once, then again after two weeks and then again after six weeks from the first scoring. We clarified the writing at p 8, ll 164-166:

“We determined the inflorescence height and the number of fully opened flowers per individual (Figure 1—figure supplement 4a). These traits were acquired thrice, in June, July, and August 2019 to account for phenological variation.”

Line 155-190. I am a bit puzzled by the floral scent collections and analyses. Usually, static collection with PMDS absorbents is used qualitatively rather than quantitatively, and in the methods it is mentioned that the compounds were not quantified but that the relative intensity of the different VOC peaks was compared. Hence, you focused on floral scent composition. This is fine, but then in the analyses of the lilac aldehydes (e.g. F), I understand it as you measure the absolute number of counts from the chromatograms.Hence, I wonder (i) were these data stilled analysed quantitatively, and, if so, how did you validate that the number of counts was comparable across samples? The traditional method is to use dynamic headspace sampling, and then add a standard volume of a known substance to account for e.g. variable GC-MS sensitivity. Alternatively, (ii) this measure is relative (as indicated by the text and the y-axis label in the figure), but then I would like a better description of how this variable was calculated. For example, if using proportions, two samples could emit the exact same amount of the particular compound, but the proportional contribution of the same compound each overall scent bouquet could be very different, if one of the two samples include large amounts also of other scent compounds. Under such a scenario, it would be misleading to use the relative contribution of one single floral scent compound to try to predict pollinator behaviour. I apologize if I have missed something obvious here, but if so, it still makes sense to clarify how these data were obtained and what may be their limitation.

We apologize because the writing definitely required clarification. Performing analyses based on compound proportions was indeed not an option for us, since we expected differences in the amount of emitted VOC among treatments that cannot be detected with these type of data. We used the described technique for a qualitative comparison (scent composition) and in addition, the total ion counts from the chromatograms were compared between samples, which we now define as “intensities”. The definition of this term has been clarified in the text p 9, ll 205-206:

“Compounds were not quantified but the intensity of the total ion chromatogram of peaks was compared among treatment groups (hereinafter referred to as intensity).”

A previous study has validated this method (Kallenbach et al. 2014, http://doi.wiley.com/10.1111/tpj.12523). Using representative plant VOC (i.e., the same chemical classes as found in this study) in a physiologically meaningful concentration range, the study has shown that the linear relationship among detected peak areas and compound concentration is comparable between VOC that have been passively trapped on PDMS (Figure S4) and the same compounds and concentrations analysed by direct liquid injection. Hence, we assume that the abundance of individual VOC in the floral headspace is correlated with the detected peak area based on passive sampling on PDMS and this justifies our comparison of intensities. We included this information into the methods section p 9, ll 207-208:

“A linear relationship among peak areas and compound concentrations has been validated for the passive sorption method in Kallenbach et al. (2014).”

It is correct that we have not added an internal standard to our individual samples. While the lack of an internal standard may cause unexplained variation in the intensity of individual VOC due to variable GC-MS sensitivity, it is highly unlikely that variable technical sensitivity caused the treatment effects illustrated in Figure 2f, as all samples were analysed in one trial in a fully randomised order, which is now stated at p 9, ll 190:

“All samples were measured in a single trial in a fully randomised order.“

Line 335-344: Again, I was confused by the term floral attractiveness, since most traits were not that important for pollinator visitation rates. This was confusing throughout the discussion.

We now avoid the term floral attractiveness throughout the manuscript and instead refer to “floral traits”.

Line 347: Also "spatial attractiveness" is a bit misleading. Why not talk about the actual traits?

We need an umbrella term for these traits for headings, tables, graphs and the discussion. “Spatial flower trait” is an umbrella term for traits describing variation in the spatial arrangement of individual flowers (e.g., size, shape, orientation, symmetry) or the spatial arrangement of multiple flowers within the inflorescence (flower number, degree of aggregation, orientation, height relative to surrounding vegetation). This term and the necessity to distinguish between spatial flower traits and flower colour traits (that were previously summarized as “visual traits”) was established in Dafny et al. 1997 “Spatial flower traits and insect spatial vision”. We clarified this definition upon first mention of the term in the introduction at p 4, ll 39-41:

“…the spatial arrangement of individual flowers (e.g., size, shape) and multiple flowers within an inflorescence (e.g., number, height above ground, degree of aggregation), i.e., spatial flower traits (Dafni et al., 1997)… ”.

Moreover, we now stick to “spatial flower traits” throughout the entire manuscript and avoided the term “spatial attractiveness”

Line 445-477: I like the discussion about how the inbreeding effects on most traits, with the potential exception of floral scent, did not result in a reduced pollinator visitation rate. I think, however, that there is room for a deeper discussion here. These other traits, are they not important for pollinator attraction at all? Or were they not reduced to a large enough extent to affect pollinator visitation?

We agree that this aspect required deeper discussion and revised the section at p 19, ll 520-526 accordingly. We believe that the limited spatial vision of *H. bicruris* in combination with our experimental setup for pollinator observations increased the relative importance of floral scent for pollinator visitation rates (suggested by referee #3).

Also, considering the impressive amount of data produced and the numerous predictor variables, how well can we trust that the floral scent reduction is indeed not just a spurious correlation? This is particularly relevant in the light of the comment above about whether or not we can be sure that you have detected variation in "aldehyde concentrations" (line 460).

As outlined in our ninth response to reviewer #2, the analyses of PDMS tubes via GC-MS were fully randomised across breeding treatments, sexes, origins and populations (so was the complete rearing of plants and the acquisition of all other data). In that way we did our best to avoid the generation of spurious correlations with treatments and random factors.

Predictors [breeding*sex*range+latitude(+age) = k = 8(9)] were tested in one model for each of the 14 responses. For sure, type 1 errors cannot be excluded here but this is rather not the range where FDR-correction is urgently required. High sample sizes rather reduce than increase the probability for type 1 errors as generated by randomly increased/decreased response values.

I was pondering whether there was an additional way to analyze these data by comparing effect sizes (trait variation between inbred and outcrossed plants of the same category) and ask how well these explained the variation in effect size in pollinator visitation rate between inbred and outcrossed plants of the same categories. I could not come up with a straightforward way to do so, so I realize the problem of drawing too strong conclusions based on a single data point (a slight reduction in aldehyde concentration in inbred females from North America and a corresponding reduced visitation rate on inbred North American females). I think that one important take-home message here is the lack of trait-based explanations for pollinator visitation rates (which further emphasizes the need to avoid using "floral attractiveness").

Indeed, this question is highly interesting and we have tried to approach this issue before submitting the manuscript to eLive initially. The straightforward solution to this question is confirmatory path analysis or structural equation modelling. With this approach one could ask which floral trait was most strongly affected by which fixed effect (sex*breeding treatment*origin) and which floral trait affected pollinator visitation rates most strongly. With both approaches one basically runs a model over a set of component models (i.e., the ones we presented in Table 2). However, for these approaches one needs exactly the same sample size and random effect structures for all floral trait and pollinator visitation models. As our sample sizes vary among floral traits (Table 1) and pollinator visitation models need to be investigated with other random effect structures (trail, plot, Table 2), we cannot use these approaches. Moreover, these analyses still cannot handle complex zero-inflation formulas, which are, however, necessary to fit appropriate component models that do not seriously violate the underlying assumptions. As such we now highlight that the relationships among pollinator visitation rates and intensities of lilac aldehydes require further investigation p 19, ll 533-534:

“However, bioassays under more controlled conditions are needed to further evaluate a mechanistic relationship among pollinator visitation and intensities of lilac aldehydes”.

Furthermore, we use the term “floral traits” instead of “floral attractiveness” throughout the revised manuscript.

Figure S8: I wonder if there is a mistake here, because the two plots in the left panel are identical.

We are sorry for this fatal mistake. We corrected the graph.

Finally, I wanted to reiterate the potential to study variation also at smaller scale. How consistent were the trait changes among populations from the same continent. Is it reasonable to lump these into the same category and to compare among continents. It would be very interesting to see some variance partitioning.

Yes, it is necessary to put populations from the same continent into one category, since native and invasive plant populations differ significantly in their evolutionary history (p 5, ll 74-81, http://onlinelibrary.wiley.com/doi/10.1111/j.1365-294X.2012.05751.x). Origin explained sufficient amounts of variation in several traits including flower number, corolla expansion, VOC diversity, lilac aldehyde A intensity, and pollinator visitation rates (see Figures 2-3; and Table 2) and some variation in in the magnitude of inbreeding effects (Figure 2e, f; Figure 3). Even if we would not be interested in differences among native and invasive populations, we would have to include origin as a fixed effect in our models because: (i) populations within a distribution range are no independent samples, (ii) origin explains sufficient variation in many responses, (iii) origin cannot be fitted as a random factor, since it has only two levels (the minimum number of levels for random effect is 4).

We agree that it would be very interesting to specifically assess differences in the magnitude of breeding and sex effects among populations within origins. We now discuss this as important future research direction at p 18, ll 500-507:

“As such, the precise mechanisms underlying variation in inbreeding effects on different scent traits across population origins of *S. latifolia* can only be explored based on comprehensive genomic resources, which are currently not available. […] This would allow a detailed quantification of geographic variation in inbreeding effects and elaborating on the causes and ecological consequences of such variation (Thompson, 2005; Schrieber and Lachmuth, 2017; Thompson et al., 2017)”.

To empirically address within-origin variation of inbreeding effects with our data, we would have to (i) fit correlated random intercepts and slopes for the interaction breeding*sex on the population random factor (models consume min. 22 DF); or (ii) include population as a fixed effect in our models (models consume min. 67 DF). We have tried both of these approaches when preparing the revision, but unfortunately it turned out that our study is not designed to address this question. The models for both variants only partially converge (see R-script ll. 1568-1580), and even if they do this does not imply that one can draw solid inference from them. Approach i often results in multiple singular convergence warning messages implying that no variance is explained by population-specific reaction norms to the fixed effects specified in the random effects structure. Approach ii results in odd rank-deficient models (I was seriously worried about type I errors here, see our thirteenth response to reviewer #2). We simply have too few replicates (5) per population*breeding treatment*sex combination for both approaches. For solid inference we would need 10_approach i_-40_approach ii_ replicates = 640-2600 individuals.

However, our experimental design is sufficient to address the hypothesis we have raised in the introduction as well as general differences in response variables among populations. We now provide information on variance partitioning for all models that include population as a random effect in S9. As you will see, population explains lower amounts of variation in our responses as the fixed effects in 9 out of 12 models. The random effects maternal and paternal genotype (mother&father) explain more variation than the random effect population in 6 of 12 cases. Thus, these data do not make a strong case for an extensive discussion of population-based differences in floral traits and this was also not a question or hypotheses we wanted to address with our study.

Reviewer #3 (Recommendations for the authors):Line 2: The first sentence of the abstract is difficult to follow, consider rephrasing. In addition, while I would definitely agree that looking at effects of inbreeding on flowers in dioecious plants is most interesting and important because of the obligate outcrossing, this study does not explore whether these effects are "particularly fatal in dioecious" as there is no comparison with a hermaphroditic or monoecious plant.

We agree that the first sentence is misleading with regard to what has been investigated and rewrote it accordingly at p 3, ll 2-4:

“We study the effects of inbreeding in a dioecious plant on its interaction with pollinating insects and test whether the magnitude of such effects is shaped by plant sex and the evolutionary histories of plant populations.”

I found the term spatial attractiveness (first encounter on line 8) quite confusing (later on also spatial floral traits). Wouldn't it simply be visual attractiveness? Of course this would not be an appropriate umbrella term considering that you also include number of flowers and inflorescence height.

We agree that the term is confusing, as we did not define it in sufficient detail in the text. “Spatial flower trait” is an umbrella term for traits describing variation in the spatial arrangement of individual flowers (e.g., size, shape, orientation, symmetry) or the spatial arrangement of multiple flowers within the inflorescence (flower number, degree of aggregation, orientation, height relative to surrounding vegetation). This term and the necessity to distinguish between spatial flower traits and flower color traits (that were previously summarized as “visual traits”) was established in Dafny et al. 1997 “Spatial flower traits and insect spatial vision”. We could think of no more appropriate term for this set of traits. We clarified this definition upon first mention of the term in the introduction at p 4, ll 39-41. Moreover, we now stick to “spatial flower traits” throughout the entire manuscript and avoided the term “spatial attractiveness”.

Considering flower number only as a component of floral attractiveness is not taking full advantage of the measure. This could also be considered an important predictor of fitness and thus quite separate from the other floral morphological traits. Same could be, to some degree, said for inflorescence height, which I assume simultaneously means plant height?In any case, I would like the authors to reconsider calling this group of floral traits "spatial".

Flower number and maybe even plant height may be employed as predictors for individual fitness (the better fitness measures would be seed output and seedling viability). However, as our article focusses on inbreeding effects on plant-pollinator interactions and not on plant fitness, we call them floral traits, which has also been done in numerous other studies on plant pollinator interactions (e.g., https://doi.org/10.1111/nph.14479).

Line 124: Are the inbred crosses between full-sibs or half-sibs? Are seeds of one capsule always sired by the same father, and if so, were the plants from each capsule kept separate? What would be the resulting F of the inbred progeny? also correct this on line 372 if they are half-sibs.

It is most likely that our experimental plants were full-sibs. However, there can be multiple paternity in *S. latifolia*, which is the reason why we cannot exclude the possibility that some family members are half-sibs and an F-value cannot be estimated. We still call it sib-mating but clearly define that this refers to either full or half-sibs at p 7, ll 134-137:

“Seeds from all maternal families (consisting of full-sibs and/or half-sibs, hereinafter referred to as sibs) were germinated and plants were grown under controlled greenhouse conditions for experimental crossings within populations”.

The plants were not kept separate in the greenhouse at the family level (in this case the family- identity effect could no longer be differentiated from the kept-at-the-same-place-in-the-greenhouse effect). The plant positions were fully randomised. We avoided uncontrolled pollination by bagging female flowers with mesh bags prior to opening. This is now described in the methods section at p 7, ll 140-142:

“During the crossings, plants were kept at randomised positions in the greenhouse. Female flower buds were covered with mesh bags prior to opening until fruit maturation and opened flowers were released from bags only for directed pollen transfer.”

Line 289: Height of inflorescence was lower? This sounds like you mean the length of the corolla.

We clarified what has been measured exactly at p 8, l 164-165:

“We determined the maximum height of synflorescences above ground level”;

p 13, ll 325-326:

“Synflorescences of inbreds had lower maximum height above ground…”; Figure 2a; and Table 1 and 2, pp 31-32.

Line 293: Perhaps it would be better to report effect sizes or percent changes here. Both differences are statistically significant, and a difference in two <0.05 p-values is not really an appropriate way of reporting a difference in the effect.

We now report percent changes calculated based on marginal estimated means.

Line 303: It is somewhat misleading to call this composition of floral VOCs when in reality you tested diversity and three individual compounds our of 70.

We agree that this formulation is misleading and changed the sentence at p 13, ll 341-342 accordingly:

“Breeding treatment, sex and population origin affected floral VOC in *S. latifolia* interactively (Table 2).”

Line 404: Why would purging only affect male flowers?

We added further details to improve clarity at pp 16-17, ll 453-459:

“Compared to females, the reproductive success of males is more limited by the availability of mates than by the availability of resources, which results in selection for increased attractiveness to pollinators (Moore and Pannell, 2011; Barrett and Hough, 2013). Competition for increased siring success among male plant individuals may create strong selection pressures that could rapidly purge deleterious recessive mutations in genes directly linked to attractiveness of male flowers to pollinators.”

Line 449: "visits were higher in inbred males." Compared to inbred females or outcrossed males?

We clarified the writing at p 18, ll 510-512.

“In North American populations, inbred females received significantly fewer plant and flower visits than outbreds, whereas flower visits were higher in inbred than outbred males (Figure 3)”.

Table 2. Ns's could be removed for clarity, a simple absence of asterisks already indicates a non-significant effect.

Done, p 32.

Add degreed of freedom.

The table caption at p 32 now states that:

“All listed fixed effects consume 1 degree of freedom.”

Moreover, DF are listed in detail in the written part of the Results section.